# Hydroquinone redox mediator enhances the photovoltaic performances of chlorophyll-based bio-inspired solar cells

Shengnan Duan[1,2,3], Chiasa Uragami[2], Kota Horiuchi[2], Kazuki Hino[2], Xiao-Feng Wang[1✉], Shin-ichi Sasaki[4,5], Hitoshi Tamiaki [5] & Hideki Hashimoto [2✉]

Chlorophyll (Chl) derivatives have recently been proposed as photoactive materials in next-generation bio-inspired solar cells, because of their natural abundance, environmental friendliness, excellent photoelectric performance, and biodegradability. However, the intrinsic excitation dynamics of Chl derivatives remain unclear. Here, we show sub-nanosecond pump–probe time-resolved absorption spectroscopy of Chl derivatives both in solution and solid film states. We observe the formation of triplet-excited states of Chl derivatives both in deoxygenated solutions and in film samples by adding all-trans-β-carotene as a triplet scavenger. In addition, radical species of the Chl derivatives in solution were identified by adding hydroquinone as a cation radical scavenger and/or anion radical donor. These radical species (either cations or anions) can become carriers in Chl-derivative-based solar cells. Remarkably, the introduction of hydroquinone to the film samples enhanced the carrier lifetimes and the power conversion efficiency of Chl-based solar cells by 20% (from pristine 1.29% to 1.55%). This enhancement is due to a charge recombination process of Chl-A$^+$/Chl-D$^-$, which is based on the natural Z-scheme process of photosynthesis.

[1] Key Laboratory of Physics and Technology for Advanced Batteries (Ministry of Education), College of Physics, Jilin University, Changchun, P. R. China. [2] Department of Applied Chemistry for Environment, Graduate School of Science and Technology, Kwansei Gakuen University, Sanda, Hyogo, Japan. [3] School of Science, Chongqing University of Posts and Telecommunications, Chongqing, P. R. China. [4] Nagahama Institute of Bio-Science and Technology, Nagahama, Shiga, Japan. [5] Graduate School of Life Sciences, Ritsumeikan University, Kusatsu, Shiga, Japan. ✉email: xf_wang@jlu.edu.cn; hideki-hassy@kwansei.ac.jp

Natural photosynthesis processes among various organisms such as plants, algae, and photosynthetic bacteria have evolved over millions of years to obtain optimized light-to-chemical energy conversion systems that sustain the lives of these organisms[1]. Artificial photosynthesis-based solar cells, which employ chlorophyll (Chl) derivatives, have been developed to simulate the energy/charge transfer of these natural photosynthesis processes[2–6]. In these systems, Chl derivatives, which have been modified from natural Chl through molecular engineering via the replacement of their peripheral functional groups, exhibit stronger light absorption, better charge separation, and better carrier transportation abilities without losing the basic photoelectric performance of natural Chls[3,7]. Owing to the increasing number of studies on various types of Chl-derivative-based bio-inspired solar cells, understanding the intrinsic properties of these Chl derivatives both in solution and thin-solid film states is essential to ensure their applicability to artificial photosynthesis-based solar cells. These bio-inspired solar cells have significant potential to achieve high photovoltaic performance by mimicking natural photosynthesis processes that have evolved over billions of years to obtain an optimized light-to-chemical conversion. In addition, these Chl-derivative-based bio-inspired solar cells are bio-degradable, which would reduce recycling costs, increase the biological compatibility, and guarantee sustainability.

Time-resolved absorption spectroscopy has been widely recognized by researchers as a useful tool for studying the excitation dynamics of materials[8–11]. Previous studies focused primarily on various types of natural Chl and light-harvesting systems without modifications, thus resulting in an insufficient understanding of Chl derivatives that are used as functional materials for solar cells[12–17]. In addition, other studies have investigated the excitation dynamics of tetrapyrrole skeleton-based porphyrin derivatives and their application in solar cells, proving that porphyrin is also a promising photosensitive material for next-generation solar cell applications[18–22]. Moreover, there are no studies in the literature that have investigated the Chl derivatives themselves using ultrafast systems and provided guidance for the next-generation solar cell design. Therefore, in this study, the solution and film samples of the Chl derivatives (see Fig. 1 for their chemical structures) zinc methyl 3-devinyl-3-hydroxymethyl-pyropheophorbide-a (Chl-A) and free-base methyl 13$^1$-deoxo-13$^1$-dicyanomethylene-pyropheophorbide-a

(Chl-D) were investigated using sub-ns pump–probe time-resolved absorption spectroscopy. Chl-A, which is a natural chlorophyll a derivative after the peripheral functional group is modified using molecular engineering to obtain a better light-to-photoelectron conversion efficiency, has been applied to organic solar cells, perovskite solar cells, organic-inorganic heterojunction solar cells, and bio-solar cells, exhibiting a significant potential to be a next-generation photoelectronic candidate[3]. Chl-D exhibits ambipolar characteristics owing to the presence of a dicynao-functional group, which makes Chl-D more attractive as both P-type and N-type semiconductors. Furthermore, the application of Chl-A and Chl-D compounds into bio-solar cells inspired by natural photosynthesis has demonstrated good performance[5,6]. However, the intrinsic excitation dynamics of Chl-A and Chl-D are still unclear[5,6,23,24]. Generally, strong light absorption, high carrier mobility, and long excitation lifetime are essential for Chl derivatives to achieve high photovoltaic performance. This is because the strong light-absorption ability can enable a device to harvest more photons, while the high carrier mobility and long excitation lifetimes contribute to charge separation and transportation[25].

In this investigation, inert nitrogen (N$_2$) gas was introduced to the solution to examine the triplet-excited states of the Chl derivatives, while all-*trans*-β-carotene was introduced to confirm their corresponding triplet states in the solid states. Specifically, these Chl derivatives in solution would have markedly increased triplet-excited lifetimes after purging with N$_2$ gas. Meanwhile, their triplet species in the films was confirmed using all-*trans*-β-carotene through the generation of triplet-excited β-carotene species originating from the energy transfer from the photo-excited Chl derivatives to β-carotene. In addition, hydroquinone (HQ), a radical cation scavenger and/or anion radical donor was employed to identify the radical species of Chl derivatives in the solutions. Either their increased or shortened lifetimes would contribute to distinguishing the radical species types (cations or anions). Surprisingly, for the film samples, we observed that the introduction of HQ to the Chl-derivative films remarkably enhanced their carrier lifetimes. This implies that the photovoltaic performance of the Chl-derivative-based bio-solar cells can be further improved through the addition of HQ to produce a longer carrier lifetime. This inference was evidenced by our improvement of the photovoltaic performance of the device after Chl-A was doped with HQ. In this study, the excitation dynamics

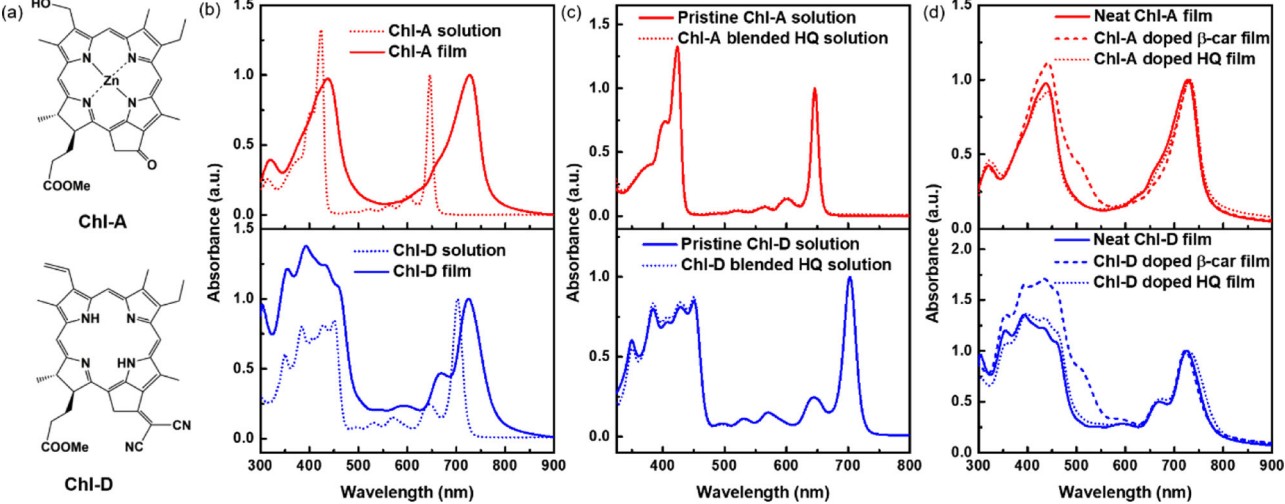

**Fig. 1 Chemical structures and steady-state absorption spectra of Chl-A and Chl-D. a** Chemical structures of Chl-A and Chl-D, (**b**) their steady-state absorption spectra in THF and in the film states, (**c**) solution samples with and without blending with HQ, and (**d**) film samples with and without blending all-*trans*-β-carotene or HQ. Note that all the data are normalized at their Q$_y$ maxima.

of Chl-A and Chl-D were investigated to facilitate further device designs for Chl-derivative-based solar cells.

## Results and discussion

Figure 1a shows the chemical structures of Chl-A and Chl-D, which were derived from the natural Chl-*a*[6]. Chl-A and Chl-D are easy to synthesize with high synthetic yields of approximately 80% and 70%, respectively[26–28]. Natural Chl-*a* was extracted from a commercial cyanobacterium, *Spirulina geitleri*. Demetallation and pyrolysis were then applied to obtain a stable intermediate. Subsequently, oxidation/reduction/metalation or dicyanomethylation changed the peripheral functional groups of the Chl macrocycle to generate Chl-A or Chl-D[3]. The absorption spectra of Chl-A and Chl-D in THF solutions and solid film states are shown in Fig. 1b. Apparent red-shifts in the film were visible compared with the absorption bands in the solution because of intermolecular interactions. In addition, the differences in the absorption spectra before and after HQ was blended were negligible for both the Chl-A and Chl-D solutions (Fig. 1c). In contrast, as shown in Fig. 1d, the differences between the absorption spectra of the all-*trans* β-carotene-blended film samples were larger than those of the HQ-blended ones, primarily because of the additional absorption from all-*trans* β-carotene.

### Sub-ns time-resolved absorption spectroscopy of Chl-A solution and film states. 

The transient absorption (TA) spectra of the pristine Chl-A solution, Chl-A solution purged with $N_2$ gas, and that blended with HQ excited at 650 nm are shown in Fig. 2a−c. Negative signals of the Chl-A samples at approximately 422 and 649 nm were visible as ground-state bleaching (GSB) signals. In contrast, the positive TA signals of the Chl-A solution were attributed to the generation of excited-state species. All the Chl-A solution samples exhibited similar TA spectra, but when these samples were purged with $N_2$ gas, their decay rates became significantly slower than that of the pristine sample (Fig. 2b).

Meanwhile, blending HQ with Chl-A in solution had a negligible effect on the TA spectra of Chl-A (Fig. 2c).

The TA spectra of the neat Chl-A film and those blended with all-*trans*-β-carotene or HQ when pumped at 737 nm are shown in Fig. 2d−f. The strong negative signals around the $Q_y$ region of the film samples were primarily caused by the scattering of the excitation laser. Therefore, the $Q_y$ bleaching signal appeared to be strong for the film samples. In addition, two bleaching signals (*Soret* and $Q_y$ bleaching) were observed for the neat Chl-A film and the film blended with HQ. The band shape and temporal change of the neat Chl-A film were similar to those of the film blended with HQ. However, the all-*trans* β-carotene-blended Chl-A film differed from the other two samples (Fig. 2e). An extra positive signal at 545 nm and a bleaching signal at 515 nm were generated for the film blended with all-*trans* β-carotene, which was attributed primarily to the TA signals of all-*trans*-β-carotene[29]. To obtain a better understanding, we applied a global analysis based on a sequential model of the entire observed dataset to the Chl-A solution and film samples. The dataset of the Chl-A solution could be adequately fitted using three components: singlet, triplet, and radical (carrier) species. Meanwhile, two components (triplet and carrier species) were used to fit the Chl-A film dataset. Here, the singlet species signal of the Chl-A film could not be properly time-resolved and was expected to be included in the instrumental response function. This originated from the shortened singlet lifetime owing to the intermolecular interaction of Chl-A in the solid film state (singlet−singlet annihilation). Figure 3a compares the evolution-associated difference absorption spectra (EADS) of the singlet species of Chl-A in solution under different conditions. The singlet species band shape of Chl-A in the solution included two bleaching signals belonging to the *Soret* and $Q_y$ bands and a broad positive TA band at 435−630 nm, which was considerably consistent with the result of the natural Chl-*a* in solution[30,31]. In comparison, the singlet lifetime of the pristine Chl-A solution was 4.1 ns, which was less than that of natural Chl-*a* at ~5 ns. This is because the reported singlet lifetime of natural Chls depends on both the

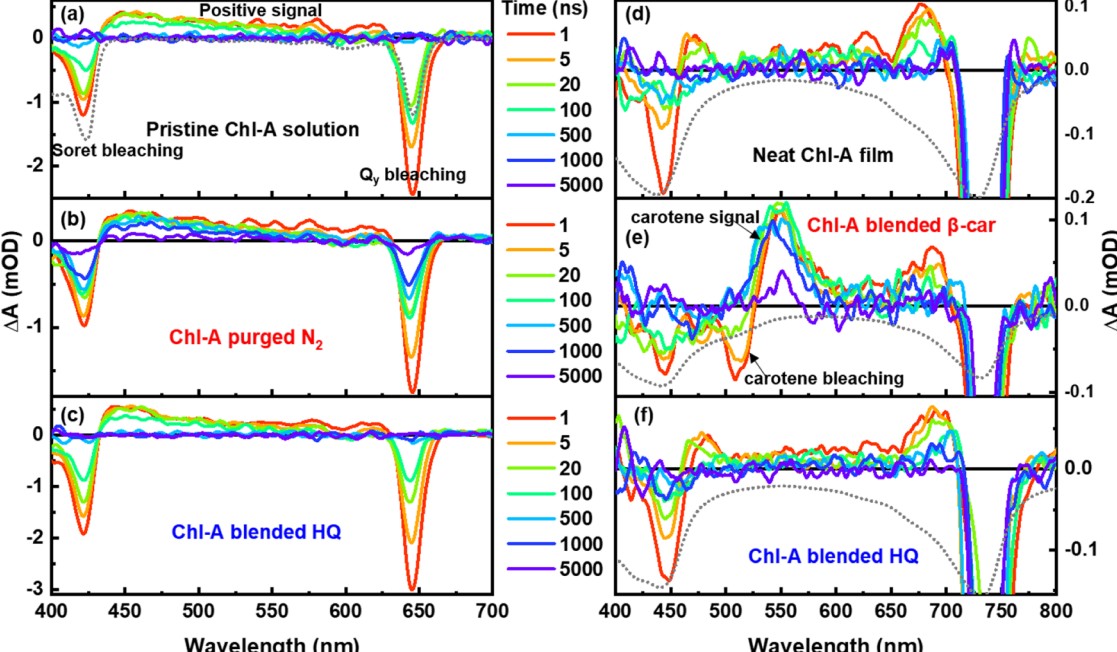

**Fig. 2 Photo-induced TA spectra of Chl-A solution and film. a** The pristine Chl-A solution and the ones (**b**) purged with $N_2$ gas or (**c**) blended with HQ excited at 650 nm ($Q_y$ band of Chl-A solution), and (**d**) a neat Chl-A film and the ones blended with (**e**) all-*trans*-β-carotene or (**f**) HQ excited at 737 nm ($Q_y$ band of Chl-A film). The gray broken-dotted line represents the inverted steady-state absorption spectrum of each sample for comparison.

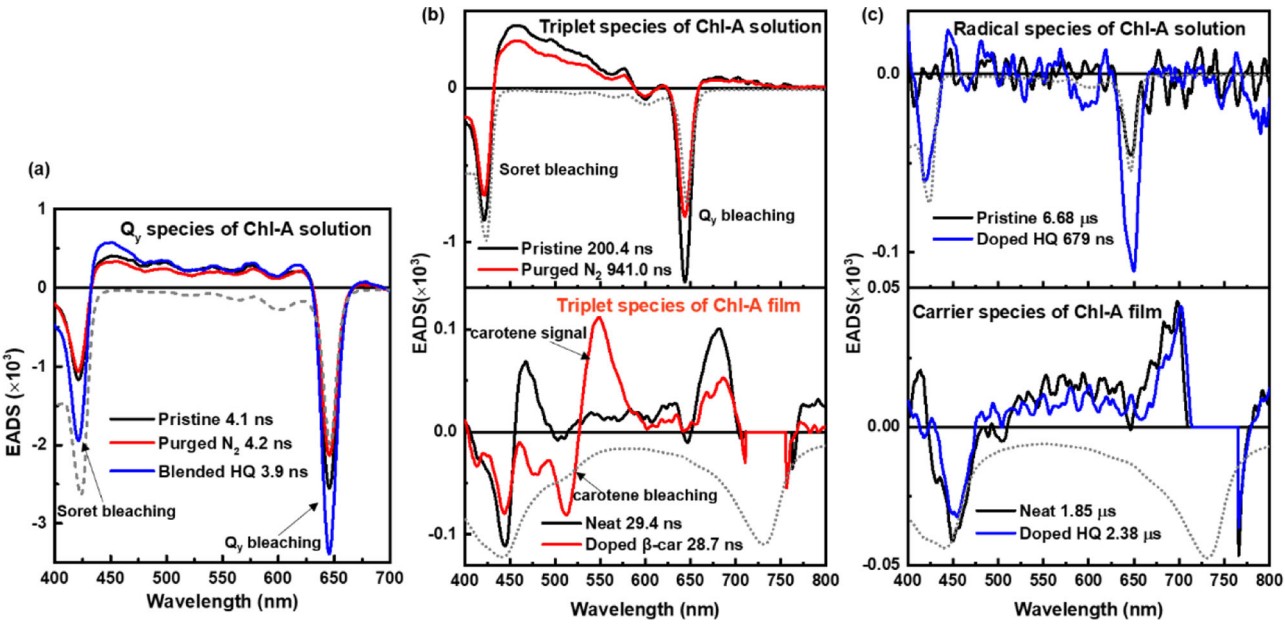

**Fig. 3 The different species of EADS comparison of the Chl-A solution and film. a** Singlet species of the pristine Chl-A solution and the ones purged with $N_2$ gas and blended with HQ. **b** EADS of the triplet species of the Chl-A solution (the pristine Chl-A and the one purged with $N_2$ gas) and Chl-A film (the neat and β-carotene-blended ones). **c** EADS of radical species of the Chl-A solution (the pristine Chl-A and the one blended with HQ) and carrier species of the Chl-A film (the neat and HQ-blended ones). (The gray broken-dotted line of each figure is the inverted steady-state absorption spectrum of those represented by colored lines.).

solvents and the differences in the molecular peripheral substituents as well as the central metal[31,32]. The $S_0$ to $S_1$ transition ($Q_y$ band) exhibits high sensitivity to solvent polarity and the π-conjugated chain. The extent of π-conjugation is closely related to the modified peripheral functional group of the tetrapyrrole ring of Chls. Furthermore, the singlet lifetime of the Chl-A solution slightly changed to 4.2 ns and 3.9 ns when the solution was purged with $N_2$ gas and blended with HQ, respectively. Similar Chl-A singlet lifetimes under different conditions indicate that $N_2$ gas and HQ had negligible effects on the singlet species of the Chl-A solution.

The triplet-excited-state species of the Chl-A solution with and without $N_2$ purging are shown at the top of Fig. 3b. The triplet lifetime of Chl-A purged with $N_2$ gas increased significantly from 200 ns (pristine Chl-A) to 941 ns. Purging the Chl-A solution with $N_2$ gas assisted in the deoxygenation of the solution. This resulted in an increased triplet lifetime, because oxygen can efficiently quench the triplet species[33,34]. Therefore, this prolonged species guaranteed the assignment of the triplet species of Chl-A. In contrast, all-*trans* β-carotene was introduced into the Chl-A film to check the triplet species of the Chl-A film. The EADS of the Chl-A film blended with all-*trans* β-carotene was composed of a mixture of Chl-A and β-carotene signals (bottom of Fig. 3b). After the film was blended with all-*trans*-β-carotene, two newly generated EADS signals appeared at 548 and 512 nm, which were not present in the neat Chl-A film. The positive signal at 548 nm was attributed to the typical triplet β-carotene species, and the negative signal at 512 nm was the bleaching signal of β-carotene. This indicated triplet−triplet excitation energy transfer from Chl-A to β-carotene. The quantum efficiency of intersystem crossing from the singlet excited state of β-carotene to the triplet state is as low as $10^{-4}$ [35,36]. However, the triplet lifetime of the Chl-A film blended with all-*trans* β-carotene remained nearly the same as that of the neat Chl-A film. This was because the Chl-A film blended with β-carotene exhibited inhomogeneity owing to the relatively low mass ratio of all-*trans*-β-carotene relative to that of Chl-A. The comparison of radical species between the

pristine Chl-A solution and that blended with HQ is shown at the top of Fig. 3c. The radical lifetime decreased from 6.68 μs (pristine Chl-A) to 679 ns after the solution was blended with HQ. As a radical cation scavenger, HQ can consume cations and cause a decrease in lifetime[37]. Therefore, this observation indicated that the Chl-A solution generated radical cations from triplet species because the successful global analysis was based on a sequential model. In contrast, the carrier-species comparison between the neat Chl-A film and the one bend with HQ is shown at the bottom of Fig. 3c. The band shapes of the neat Chl-A film and the HQ-blended film were similar; however, the carrier lifetime increased surprisingly from 1.85 μs (neat Chl-A film) to 2.38 μs (Chl-A blended with HQ). This could be attributed to the longer triplet lifetime of Chl-A blended with HQ (40.7 ns) than that of the neat states (29.3 ns) (Fig. S1), considering that the carrier (radical species) was generated from the triplet species. This was a unique finding in this study, which indicated that the HQ (a redox mediator) can be applied to Chl-based solar cells to improve their photovoltaic performance.

**Sub-ns time-resolved absorption spectroscopy of Chl-D solution and film states.** The TA spectra of the pristine Chl-D solution and those purged with $N_2$ gas or blended with HQ excited at 704 nm are shown in Fig. 4a−c. The negative TA signals at the $Q_y$ and *Soret* bands were included among the GSB signals. The positive absorption changes from 460 to 680 nm were attributed to the generation of excited-state species. There was a pronounced retardation in the TA signal change when the solution was purged with $N_2$ gas (Fig. 4b). This suggested the presence of long-lived transient species after the solution was purged with $N_2$ gas. For the HQ-blended Chl-D solution, its TA decay signals exhibited behavior similar to that of pristine Chl-D over the entire timescale (Fig. 4c), which is in contrast to the TA spectra of the Chl-D film shown in Fig. 4d−f. Three bleaching signals at the *Soret*, $Q_x$, and $Q_y$ maxima were observed for the neat Chl-D film (Fig. 4d) and the one blended with HQ (Fig. 4f).

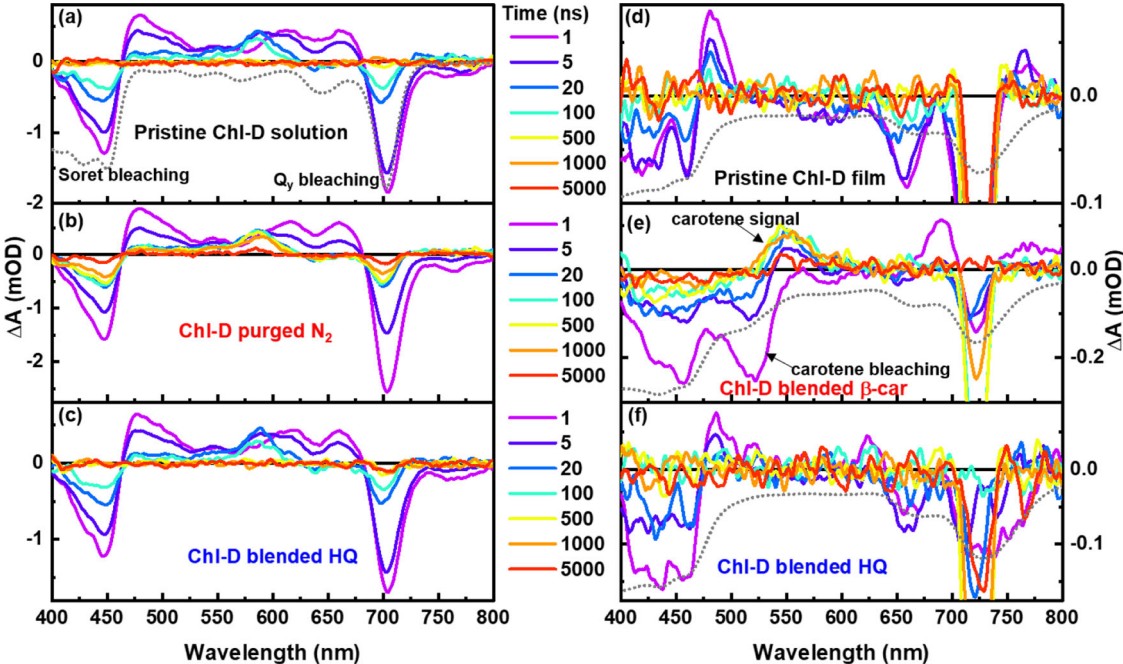

**Fig. 4 Photo-induced TA spectra of Chl-D solution and film. a** Pristine Chl-D solution and the ones (**b**) purged with $N_2$ gas or (**c**) blended with HQ excited at 650 nm ($Q_y$ band of Chl-D solution), and the TA spectra of Chl-D film which included (**d**) neat Chl-D film and the ones blended with (**e**) β-carotene or (**f**) HQ excited at 737 nm ($Q_y$ band of Chl-D film). The gray broken-dotted line represents the inverted steady-state absorption spectrum of each sample for comparison.

Meanwhile, the TA decay signal of the Chl-D film blended with all-*trans*-β-carotene (Fig. 4e) exhibited an apparent difference in that one more bleaching signal was generated at 523 nm, which was attributed to all-*trans*-β-carotene. In addition, a rapid positive signal at 685 nm was generated for Chl-D blended with all-*trans*-β-carotene. This signal disappeared rapidly together with the generation of a new positive signal at 545 nm, which is a typical β-carotene triplet signal.

For the Chl-D solution, three components (singlet, triplet, and radical species) were used to fit the Chl-D solution dataset, while two components (triplet and radical (carrier) species) were used to fit the Chl-D film dataset. Both the band shape and lifetime of the singlet species of the Chl-D solution under different conditions remained similar (Fig. 5a). For the triplet species, the lifetime of the $N_2$ purged Chl-D solution was prolonged significantly from the pristine 174.8 ns to the present 4.1 μs because of the deoxygenation of the Chl-D solution (Fig. 5b)[38]. In contrast, the triplet-species comparison of the Chl-D film with and without all-*trans*-β-carotene is shown at the bottom of Fig. 5b. The original triplet Chl-D positive signal at 480 nm for the Chl-D film blended with all-*trans*-β-carotene disappeared together with the generation of a new positive signal at 550 nm. This new positive signal at 550 nm is a typical triplet β-carotene signal owing to the energy transfer from Chl-D to β-carotene.

Figure 5c shows a comparison of the radical (carrier) species of the Chl-D solution (or film) with and without HQ. The radical lifetime of Chl-D solution after being blended with HQ is increased from the pristine 3.3 μs to the present 14.2 μs, which was significantly different from that of Chl-A. This increased lifetime of the Chl-D solution blended with HQ indicated that the Chl-D radical species is an anion. More specifically, the Chl-D anion received electrons from the HQ, causing larger relaxation time from Chl-D²⁻ to Chl-D than from Chl-D⁻ to Chl-D. Therefore, the radical lifetime of the Chl-D solution after blending with HQ was longer than that of the pristine one. In contrast, the carrier-

**Table 1 Comparison of the triplet and carrier lifetimes in the Chl-A and Chl-D films with and without HQ.**

|  | Chl-A | | Chl-D | |
|---|---|---|---|---|
|  | **Without HQ** | **With HQ** | **Without HQ** | **With HQ** |
| Triplet lifetime (ns) | 29.3 | 40.7 (×1.39 of w/o HQ) | 22.6 | 42.3 (×1.87 of w/o HQ) |
| Carrier lifetime (μs) | 1.85 | 2.38 (×1.28 of w/o HQ) | 0.653 | 9.05 (×13.9 of w/o HQ) |

species comparison of the Chl-D film with and without HQ is shown at the bottom of Fig. 5c. Here, the band shapes of these two types of carrier species were similar, but their lifetimes differed significantly. This may be explained by two factors. First, it may be attributed to the increment in the triplet lifetime from 22.56 ns (neat Chl-D film) to 42.30 ns (Chl-D film blended with HQ), thus resulting in a longer carrier lifetime (Fig. S2). Second, the electron-donating function of HQ to the Chl-D anion enlarged the relaxation time from pristine Chl-D⁻ → Chl-D to Chl-D²⁻ → Chl-D. The 1.87 times increase in the triplet lifetime of the Chl-D film did not directly correspond to the 13.9 times increase in the lifetime of the charged carrier species. In contrast, for the Chl-A film, the rate of elongation of the carrier lifetime was proportional to that of the triplet lifetime (Table 1). Therefore, a more than 10 times increase in the carrier lifetime for the Chl-D film should be caused by a carrier generation mechanism different from that of the Chl-A film. As exemplified in the solution samples, HQ acted as a cation scavenger for Chl-A. This indicated that HQ is an electron donor to Chl-A. For the Chl-D solution, the radical species were assigned to the anion in this study, and HQ donated additional electrons to Chl-D⁻ to generate Chl-D²⁻. Therefore,

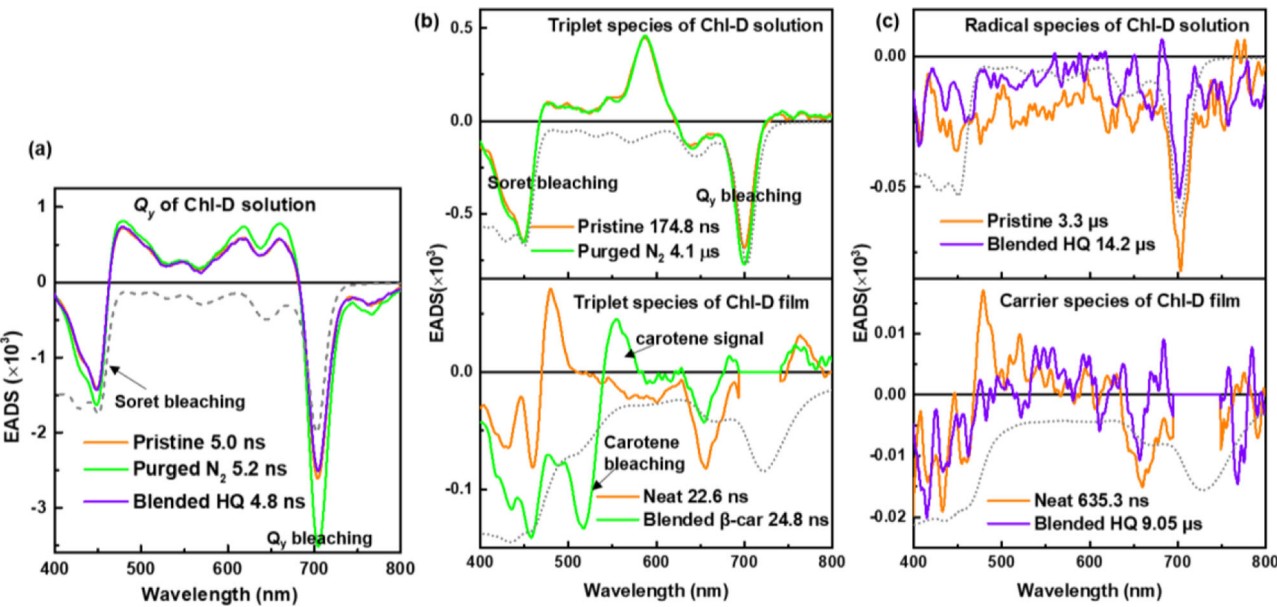

**Fig. 5 The different species of EADS comparison of the Chl-D solution and film. a** EADS of singlet species comparison of the pristine Chl-D solution and the ones purged with N₂ gas or blended with HQ. **b** EADS of triplet species comparison of the Chl-D solution (the pristine and N₂ purged ones) and Chl-D film (the neat and β-carotene-blended ones), and (**c**) EADS of the radical species comparison of the Chl-D solution (the pristine one and the HQ-blended one) and carrier species of Chl-D film (the film with and without blending with HQ). The gray broken-dotted line in each figure is the inverted steady-state absorption spectrum of each sample for comparison.

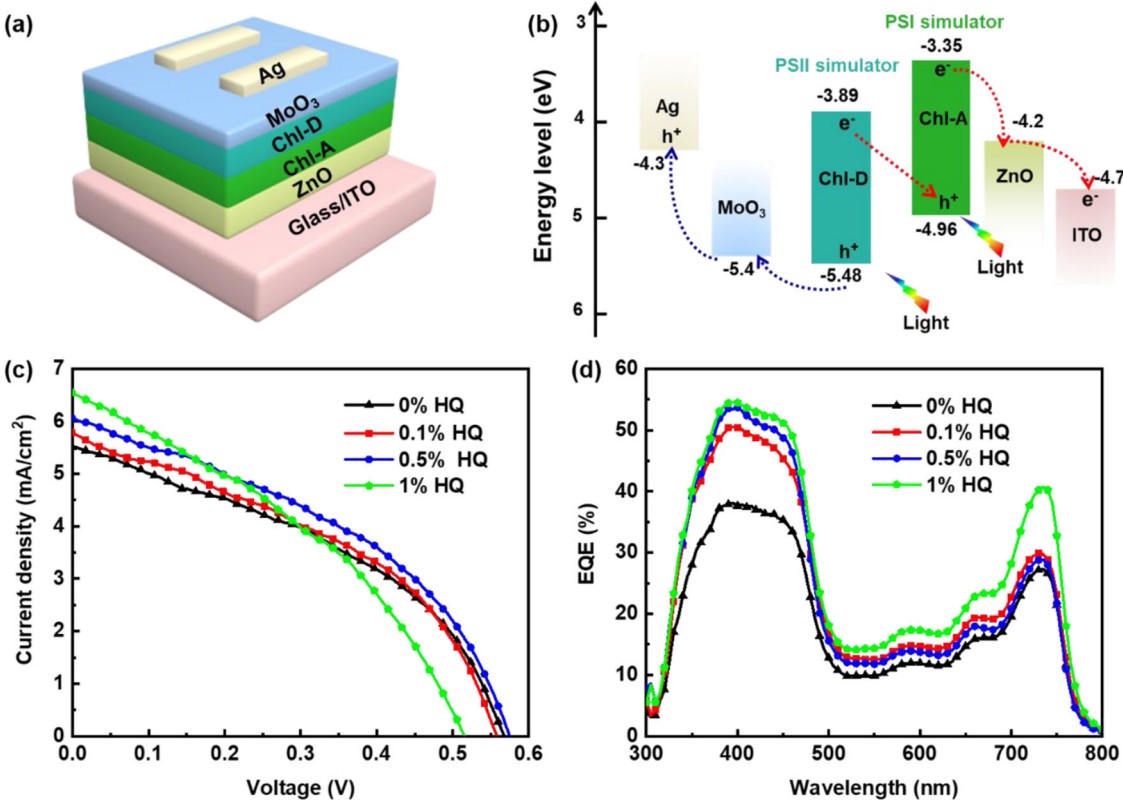

**Fig. 6 Photovoltaic performance related parameters of the Z-scheme process of oxygenic photosynthesis inspired bio-solar cells. a** Device architecture, (**b**) energy alignments, (**c**) J−V curves, and (**d**) EQE of the bio-solar cells with different HQ doping ratios to Chl-A layer.

this may be the reason for observing a carrier lifetime elongation of more than 10 times for the Chl-D film. If we accept this concept, another question may arise: Why did the carrier lifetime in the Chl-A film increase upon doping with HQ, since HQ should act as a cation scavenger? Here, we must assume again that the elongation of the triplet lifetime in the Chl-A film by doping HQ should be the main reason for the elongation of the carrier lifetime. This concept can be supported by considering the rate of elongation of the triple and carrier lifetimes by HQ doping (Table 1). The rate of elongation for the carrier lifetime (×1.28)

**Table 2 Photovoltaic performances of the Chl-derivative-based bio-solar cells with different HQ doping ratios.**

| Device types | $J_{sc}$ (mA cm$^{-2}$) | $V_{oc}$ (V) | FF | PCE (%) |
|---|---|---|---|---|
| 0% HQ | 5.54 | 0.57 | 0.41 | 1.29 |
| 0.1% HQ | 5.83 | 0.56 | 0.42 | 1.37 |
| 0.5% HQ | 6.09 | 0.58 | 0.44 | 1.55 |
| 1% HQ | 6.62 | 0.52 | 0.38 | 1.31 |

was slightly smaller than that of the triplet species (×1.39), which might have been due to the cation scavenging activity of HQ in the Chl-A film.

**The photovoltaic performance enhancement of the Chl-A and Chl-D inspired bio-solar cells**. To check whether the increased carrier lifetime (by doping with HQ) could be attributed to the generation of the photocurrent, the Z-scheme process of oxygenic photosynthesis simulated bio-solar cells upon doping with HQ were fabricated as shown in Fig. 6a, and their corresponding energy alignments are shown in Fig. 6b. The corresponding photovoltaic parameters are summarized in Table 2. Chl-A and Chl-D act as light-harvesting materials in the 350–700 nm wavelength region, which resembles the Z-scheme of natural photosynthesis. Therefore, we called our device a bio-inspired solar cell. The incident photons of Chl-A and Chl-D generate photo-excited carriers. A cation radical (hole) is produced in the Chl-A layer, and the electron is transferred to the cathode, while the anion radical (electron) is produced in the Chl-D layer, and the electron is transferred to the anode. These carriers are recombined at the interface between the Chl-A and Chl-D layers, and a photocurrent is finally generated by this device under illumination. The device optimization process for the film thickness is shown in Fig. S3. The Chl-A layer was doped with HQ at ratios of 0.1%, 0.5%, and 1%. The results demonstrated that the 0.5% HQ-doped device achieved the highest power conversion efficiency (PCE) of 1.55%, which was larger than that of the pristine Chl-A-based device (1.29%). This was because the photocurrent and fill factor (FF) of the 0.5% HQ-doped device increased. In addition, the photocurrent of the device increased as the HQ ratio increased, which was consistent with the results of our EADS analysis (i.e., the participation of HQ could prolong the carrier lifetime of Chl-A). However, the voltage and FF could be reduced rapidly if the HQ ratio was high. This was because the charge recombination in the device became severe by doping a large ratio of dopant[39].

We also conducted external quantum efficiency (EQE) measurements of the devices with different HQ doping ratios, and the results are shown in Fig. 6b. The EQE response of each device was consistent with the photocurrent calculated from the $J-V$ curves. In addition, bio-solar cells with HQ doped into Chl-A and/or Chl-D layers at a ratio of 1:1 were fabricated, but the photovoltaic performances of these devices were worse than that of the pristine Chl-D-based device (Fig. S4 and Table S1). This was because the overhigh HQ ratio against the Chl-A or Chl-D film may have interfered with the self-assembly aggregation of the Chls. Thus, we reduced the ratio of HQ and observed a positive function when HQ was blended with Chl-A at a ratio of 0.5%. The HQ-blended Chl-D-based device exhibited a much lower performance. The devices were also doped with different HQ ratios to the Chl-D layer (Fig. S5 and Table S2). The results always indicated worse performance when HQ was doped into the Chl-D layer. The photovoltaic performance of the device decreased with an increase in the HQ ratio, which proved that

HQ may not always be functional to any Chl-derivative-based bio-solar cells by doping.

We consider that our solar cell is not a traditional P−N heterojunction-type semiconductor device. In this study, we confirmed that the photo-excited carriers generated in the Chl-A film are cations (holes) and those in Chl-D films are anions (electrons). These carriers are generated at each layer (not at the interface between the Chl-A and Chl-D layers). Therefore, in our solar cell, the electrode adjacent to the Chl-A layer collects electrons, which are discharged from Chl-A, and that adjacent to the Chl-D layer collects holes, which are discharged from Chl-D. While in the typical P−N heterojunction devices, the directions of the carrier flows are opposite. The electrode that is connected to the P-type semiconductor collects holes and that connected to the N-type semiconductor collects electrons. We consider this as a unique feature of our device.

For our device to form an electric circuit, the population of holes in the Chl-A layer and that of electrons in the Chl-D layer should be equilibrated. A balanced carrier lifetime between the donor and acceptor prevents unnecessary charge build-up and consequently reduces the probability of useless charge recombination in either the Chl-A or Chl-D layers. In other words, in charge-unbalanced cells with significant differences in hole and electron carrier lifetimes, the rapping of the slow carriers results in a build-up of space charges, which induces the reduction of photocurrents by unwanted recombination of the carriers. Therefore, we consider that the device performance might deteriorate when the carrier lifetimes of the donor and acceptor are significantly different. Therefore, balanced donor and acceptor transportation contributes to achieving optimized photovoltaic performance, among which balanced donor and acceptor carrier lifetimes have a central function. The carrier lifetime of HQ-doped Chl-D (9.05 μs) is much higher than those of Chl-A, i.e., 2.38 μs (HQ-doped Chl-A) and 1.85 μs (pristine). Consequently, a worse photovoltaic performance can be observed for the device based on Chl-D doped with HQ than that of the pristine device without doping. In addition, the photovoltaic performance comparison between the forward and backward scanning directions of the 0.5% HQ-doped device is shown in Fig. S6, and the differences between different scanning directions are negligible.

The aim of this study was to clarify the excitation dynamics of Chl-A and Chl-D both in the solution and film states, and consequently determine the operating principle of the Chl-A and Chl-D based natural Z-scheme photosynthesis inspired bio-solar cells. After this investigation, we concluded that our hypothesis is correct. We considered it necessary to propose the exact operating mechanisms of the relevant bio-solar cell device with our finding, therefore, we combined the device energy alignments of each layer and the excitation dynamics of the Chls, and the explanation is depicted in Fig. 7.

Electron and hole pairs (excitons) are generated from triplet Chl-A and Chl-D after photoexcitation. The generated holes of Chl-A are recombined with the generated electrons of Chl-D at the interface of the Chl-A and Chl-D layers. In contrast, the electrons discharged from Chl-A are extracted by the ZnO electron transporter and collected by the indium tin oxide (ITO) cathode, while the holes discharged from Chl-D migrate to the Ag anode through the MoO$_3$ electron blocker when the device is operating. This is consistent with the result of the natural Z-scheme process of the oxygenic-photosynthesis-inspired bio-solar cell, for which there was an important but unconfirmed assumption in the previous work (i.e., there is a charge recombination between Chl-A and Chl-D). The results of this study clearly demonstrate that the electrons of Chl-D would be recombined with the holes of Chl-A. Therefore, we can conclude

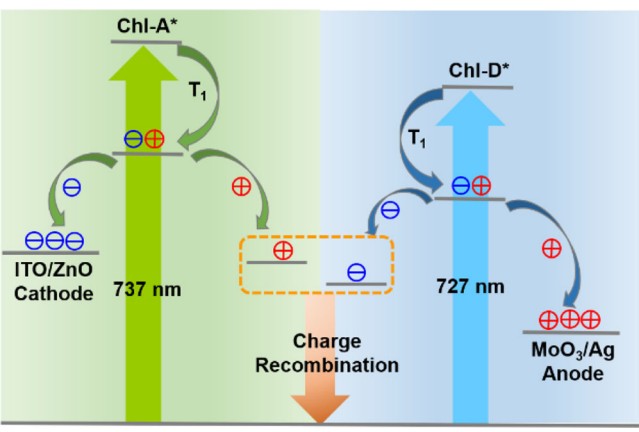

**Fig. 7 Analytic model used for the Chl-A and Chl-D inspired bio-solar cell.** Model used for the Chl-A and Chl-D excitation dynamics, charge transfer, and recombination processes occurring in the Z-scheme process of an oxygenic photosynthesis inspired bio-solar cell.

that the following reaction occurs:

$$Chl - A^+ + Chl - D^- \rightarrow Chl - A + Chl - D,$$

which express the performance of the bio-solar cells.

## Conclusion
In this study, we determined the excited-state species of Chl derivatives both in solution and film states using sub-ns TA spectroscopy, which would contribute to solar cell designs that use naturally abundant, toxic-free, and low-cost Chl derivatives. Additionally, this study demonstrated a longer carrier lifetime for both Chl-A and Chl-D films blended with HQ, providing a simple method of enhancing the performance of current Chl-derivative-based bio-solar cells. The advanced photovoltaic performance of the device after the Chl-A layer was doped with HQ strongly supports our conclusion. This study clearly demonstrated the importance of understanding the basic characteristics of photo-excited carrier species in designing Chl-derivative-based bio-solar cells.

## Experimental section
**Sample preparation.** Chl-A[40] and Chl-D[41] were synthesized according to reported procedures and dissolved in tetrahydrofuran (THF) solvent to achieve an absorbance of 0.3 at their $Q_y$ maxima when the solution was placed in a cuvette with a 2 mm optical path length. All the solution samples were stirred during the measurements using a magnetic stirrer bar to avoid optical damage during the measurements. For the $N_2$ purged samples, there was a 30 min preprocess of $N_2$ bubbling, and this process lasted for the entire measurement time at a speed of 1 ml/ min. In contrast, 1.1 mg/ml HQ was dissolved in THF to prepare the HQ-blended solution according to a previous study[42]. Subsequently, Chl-A or Chl-D was dissolved in the HQ-blended solution to obtain an absorbance of 0.3 at their $Q_y$ maxima in the same cuvette. No $N_2$ bubbling or HQ blending was conducted for the pristine Chl-A and Chl-D solution samples.

For the film samples, Chl-A was dissolved in THF solution at a concentration of 12 mg/ml, while Chl-D was dissolved in chloroform at the same concentration. For all-*trans* β-carotene-blended film samples, the molar ratio of Chl-A (or Chl-D) to β-carotene was 5:1, while that of Chl-A (or Chl-D) to HQ was 1:1. According to the Beer−Lambert law, the molar ratio of HQ to Chl-A in the film sample was estimated to be approximately 1000 times larger than that in the solution sample. Since the electron-

transfer process in the solution samples was diffusion-limited, a large amount of HQ was required to detect its radical scavenging ability. However, Chl-A and HQ were fixed to each other in the film; here, electron transfer between HQ and Chl-A became easier. Therefore, the optimized HQ:Chl-A molar ratio was 1:1 for the HQ-blended film samples to record the meaningful TA signal changes induced by the presence of HQ. All the samples were spin-coated at a speed of 1300 rpm (the thickness of the films was estimated to be 47 nm). Since our experiment and analysis were based on a target model, adequate ratios of HQ and all-*trans* β-carotene were applied to the film samples to capture their TA signals.

**Device fabrication.** The Z-scheme process of oxygenic-photosynthesis-inspired bio-solar cells were fabricated as ITO/ZnO/Chl-A/Chl-D/MoO$_3$/Ag. The ITO substrate was precleaned twice for 30 min in an ultrasonic bath using the following sequence: ITO glass detergent, deionized water, ethanol, acetone, and isopropanol. A ZnO precursor solution was prepared by mixing 0.2 g of zinc acetate dihydrate into 2 ml of 2-methoxyethanol and 56 μl of ethanolamine was included as a stabilizer, followed by stirring overnight. Chl-A was dissolved in THF at a concentration of 6.5 mg/ml and HQ was doped at ratios of 0.1%, 0.5%, and 1%. Chl-D was dissolved in chloroform at a concentration of 6 mg/ml. The cleaned ITO substrate was exposed to ultraviolet ozone for 30 min. Subsequently, the ZnO precursor solution was spin-coated onto the substrate at a spin speed of 4000 rpm and annealed immediately at 200 °C for 30 min in air. Thereafter, the substrate was transferred into a glove box, and the HQ-doped Chl-A solution was spin-coated onto the ZnO substrate at a spin speed of 2800 rpm, followed by spin-coating the Chl-D solution at a speed of 2800 rpm. The thicknesses of the Chl-A and Chl-D layers were estimated to be 22 nm based on cross-sectional scanning electron microscopy measurements of the pristine Chl-A film. Subsequently, the substrate was transferred into a high-vacuum chamber where MoO$_3$ (10 nm) and Ag (100 nm) were thermally deposited in an orderly manner.

**Steady-state absorption and time-resolved absorption measurements.** Steady-state absorption spectral measurements were performed using a JASCO V-670 UV-vis spectrophotometer. Sub-ns pump–probe spectroscopic measurements were performed using a combination of a Ti:Sapphire regenerative amplifier (Hurricane-X, Spectra Physics, 100 fs pulse duration at 800 nm) and a sub-ns pump–probe time-resolved absorption spectrophotometry system (EOS Vis-Nir, Ultrafast Systems)[30,43]. Excitation pulses at 650 nm (737 nm) for Chl-A solution (or film) and 704 nm (727 nm) for the Chl-D solution (or film) were prepared through the second harmonic generation of the output of an optical parametric amplifier (OPA-800 CF, Spectra Physics). The excitation intensity was set to 20 nJ/pulse for the solution samples and 50 nJ/pulse for the film samples with a pulse width of 100 fs and a beam diameter of 200 μm. The time delay between the pump and probe pulses was adjusted electronically (Ultrafast Systems, EOS). All the measurements were performed at room temperature. The film samples were fixed onto a Lissajous scanner during the measurement. This scanner could move the sample film according to the Lissajous curve to avoid long-time exposure of the laser light onto the same spot of the film. The instrumental response function of the system was estimated to be approximately 500 ps, which corresponded to the pulse width of the probe super-continuum light; however, the pump light was as short as 100 fs. Global and target analyses were applied to all the observed datasets of sub-ns time-resolved absorption spectra using a Glotaran program[44].

**Device characterization**. The photocurrent density–voltage ($J$ $-V$) characteristics of the solar cells were determined using a computer-controlled Keithley 2400 source meter measurement system with an AM1.5 G filter at a calibrated intensity of 100 mW/cm$^2$ illumination, as corrected by a standard silicon reference cell (91150 V Oriel Instruments). The scan direction was the forward type in the main test at a speed of 140 mV/s. This test was conducted under air atmosphere without any preconditioning of the device. Incident photon-to-electron conversion efficiency (IPCE) measurements were conducted using a commercial IPCE setup equipped with a 100 W Xe arc lamp, filter wheel, and monochromator (Crowntech QTest Station 1000AD, SOFN Instruments CO., Ltd.). Monochromatic light chopped at a frequency of 80 Hz was used to irradiate the cells, and photocurrents were measured using a lock-in amplifier. The device area (0.04 cm$^2$) was controlled using a metal mask. The final performance of each device was averaged over 10 independent solar cells.

**Reporting summary**. Further information on experimental design is available in the Nature Research Reporting Summary linked to this paper.

## Data availability

The data that support the findings of this study are available from the authors on reasonable request.

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

## Acknowledgements

The authors gratefully acknowledge the financial support from the China Scholarship Council. This work was supported by the National Natural Science Foundation of China (No. 11974129) (to X.-F.W.), the Fundamental Research Funds for the Central Universities, Jilin University, JSPS KAKENHI in Grant-in-Aids for Basic Research (B) (No. 16H04181) (to H.H.), and JSPS KAKENHI in Scientific Research on Innovative Areas "Innovation for Light-Energy Conversion ($I^4$LEC)" (Nos. 17H06433 and 17H06437) (to H.H.) and (No. 17H06436) (to H.T.)

## Author contributions

S.D. made all the samples for the TA measurements and analyzed the TA data as well as fabricated the solar cells and wrote the manuscript. C.U. manipulated the TA measurement and analyzed the data. K. Horiuchi and K. Hino contributed to set up the TA spectroscopy system. H.H. and X.-F.W. supervised the work of S.D., C.U., K. Horiuchi, and K. Hino. S.S. and H.T. synthesized Chl-A and Chl-D. All authors contributed to revise the manuscript.

## Competing interests

The authors declare no competing interests.
