## [Peer Review File · Communications Chemistry]

Reviewers' comments:

Reviewer #1 (Remarks to the Author):

The manuscript "Redox mediator for enhancing the photovoltaic performance of chlorophyll-based bio-inspired solar cells" by Shengnan Duan et al. covers the construction of bio-solar cells and evaluation of their performance through the analysis of chlorophyll derivatives including Chl-A and Chl-D. In particular, the excitation dynamics were investigated in the film and solution forms of Chl-A and Chl-D, and in the process, hydroquinone(HQ), which is a cation radical scavenger and/or anion radical donor, and all-trans- β -carotene, which is a triplet scavenger, were added to measure the life time of radical species and carrier species. Based on the results of the life time of radical and carrier species, bio-inspired solar cells with different HQ ratios to the Chl-A were constructed and verified. Overall, the process of analyzing chlorophyll derivatives, constructing cells, and evaluating their performance was well organized, but with some lack of explanation, some questions still exist. For these reasons, the manuscript cannot be acceptable in the present form. I would suggest the authors consider the points raised below.

Major issues

1. It is only mentioned that Chl-A and Chl-D were used in previous studies. Since these chlorophylls are substances that are mainly analyzed in this study, it should be explained in the introduction specifically what advantages they are used for and what characteristics they have.
2. In the experiment section, the preparation processes of Chl-A and Chl-D solution samples and film samples are different. (a) Why the concentration or amount of Chlorophyll used is not the same? (b) Why are the HQ ratios different? (c) What is the thickness of the spin-coated film samples at 1300 rpm? (d) Why are the device samples spin-coated at 2800 rpm? (e) The HQ ratio of the device samples is very low compared to that of the film samples. Is there a result of film samples with a low HQ ratio or device samples with a high HQ ratio?
3. Regarding Figure 2, it is necessary to explain the reason or mechanism in which the negative signal does not appear above 500 ns.
4. (line 23-24, page 9) The authors mentioned "singlet lifetime of natural Chls depend on both the solvents and the differences in the molecular peripheral substituents and central metal." The evidence of this statement should be explained in the text, as well as the papers referenced by authors should be cited.
5. (line 3-8, page 15 and figure 5 (c)) The lifetime of carrier species of Chl-D film with or without HQ blending differs more than 10 times, but the triplet lifetime (figure S2) is only doubled. What makes this difference?
6. (Figure 6) Why does the voltage of the device decrease with the higher HQ ratio than 0.5%?
7. (line 8-16, page 17) Why is the life time balance of donor and acceptor important? And why does the performance deteriorate when they are different? In addition, if the HQ concentration in the Chl-D film samples is very high and the life time balance is not suitable, it is expected that the life time balance of the device sample can be fitted by adjusting the HQ concentration of Chl-D. The authors should provide the results of a life time balance with Chl-A by adjusting the HQ concentration of Chl-D. Otherwise, it is not convincing that HQ doping of Chl-D simply degrades

performance.

8. The authors asserted that adopting HQ can improve the performance of bio-solar cell performance by 20%. However, considering the power conversion efficiency, it does not appear to be significantly improved (from 1.29% to 1.55). The novelty of this paper can be improved by comparing/analyzing the results with other previous studies.

Minor issues

1. Why was all-trans- β -carotene not investigated in chlorophyll solution?
2. In Figures 2 to 5, it is recommended to indicate what each peak is related to in the graph.

Reviewer #2 (Remarks to the Author):

The manuscript entitled, "Redox mediator for enhancing the photovoltaic performance of chlorophyll-based bio-inspired solar cells" reports the influences of redox mediator on bio-inspired solar cell performances. This research provides salient insights in both the characterization and application. Hence, this manuscript can be considered for publication after the mandatory revision on the given aspects :

1. Language of the manuscript should be improved.
2. The introduction part should be precise. So far numerous efforts have been reported on the artificial photosynthesis-based solar cells. In this context, it is important to draw clear lines between other close works and the present work.
3. The authors should depict the advantages of bio-inspired solar cells over the conventional solar cells.
4. The synthesis advantages of chl-A and chl-D should be illustrated.
5. The significant role of Chl-derivatives on power generation should be detailed.
6. The reasons for high power generation of a specific chl derivative should be depicted.
7. Have the authors optimized thickness of variant layers in solar cell fabrication? If so, detail the same.
8. The mechanism involved in the power generation should be schematically illustrated.

Reviewer #3 (Remarks to the Author):

The authors provide compelling evidence that chlorophyll A and D have long lived triplet states that are quenched by O₂ in solution and in thin films. Many macrocyclic organic excited states, such as porphyrins, show similar behavior. This reviewer does not know how novel this observation is for chlorophylls.

The triplet states are competent of oxidation of hydroquinone (HQ) and/or reduction of beta-carotene. Energy transfer to beta-carotene was also observed. It was unclear whether the authors believe that the triplets annihilate (i.e. disproportionate) and then the products are scavenged or whether the triplets undergo direct electron transfer with HQ and B-car. The 'lifetimes' are longer after these electron transfer events than for the triplets alone.

Solar cells comprised of these materials show $> 5 \text{ mA/cm}^2$ of current density and monochromatic EQEs $> 50\%$ which is quite good.

Overall I found this paper to be confusing to read. Calling the HQ and B-Car scavengers and referring to their charge rather than calling them acceptors and donors and referring to whether they were reduced and oxidized was unfortunate. Figure 7 looks more like a diagram for a semiconductor than for molecular dyes. Also referring to 'carriers' was vague as in some cases this was referring to chlorophyll triplets and in other cases charge-separated states. If there is a lot of triple-triple annihilation then the charge separated states will likely live longer and this is not a surprise. The true lifetime of organic triplets remains contentious in many cases as they are so long-lived that their lifetimes are often limited by quenching with impurities.

Response to the editor's comments:

As you will see, all Reviewers mention that the manuscript would benefit from a clearer presentation of the advance your work represents over closely related prior work. In addition, Reviewer #1 notes that the choice of chlorophylls and the preparation of solution and thin films is currently unclear, and that the device performance such as voltage decrease need to be clarified. The claim that HQ doping of Chl-D degrades performance needs to be further substantiated with data.

Reply: We have made substantial revisions to the manuscript. The revised parts are in blue text color in the revised manuscript.

(1) We emphasized the importance of this study and its differences from the previous studies. The corresponding revisions are provided in the Introduction of the manuscript.

(2) The choice of Chls and the preparation of the solution and thin films, which Reviewer 1 mentioned, have been answered specifically in the main text (please see the reply to Reviewer 1).

(3) The reason for the changes in the device performance after HQ doping with different ratios has been explained using the data obtained from additional experiments.

(4) The photovoltaic performances of the HQ-doped Chl-D devices have been included and discussed in the manuscript, and the relevant results are depicted in Figure S5 of the Supporting Information.

Responses to Reviewers' comments:

Responses to the comments from Reviewer #1:

The manuscript "Redox mediator for enhancing the photovoltaic performance of chlorophyll-based bio-inspired solar cells" by Shengnan Duan et al. covers the construction of bio-solar cells and evaluation of their performance through the analysis of chlorophyll derivatives including Chl-A and Chl-D. In particular, the excitation dynamics were investigated in the film and solution forms of Chl-A and Chl-D, and in the process, hydroquinone (HQ), which is a cation radical scavenger and/or anion

radical donor, and all-trans- β -carotene, which is a triplet scavenger, were added to measure the life time of radical species and carrier species. Based on the results of the life time of radical and carrier species, bio-inspired solar cells with different HQ ratios to the Chl-A were constructed and verified. Overall, the process of analyzing chlorophyll derivatives, constructing cells, and evaluating their performance was well organized, but with some lack of explanation, some questions still exist. For these reasons, the manuscript cannot be acceptable in the present form. I would suggest the authors consider the points raised below.

Reply: Thank you very much for your positive evaluation of our study. We have made revisions according to your suggestions.

Major issues

1. It is only mentioned that Chl-A and Chl-D were used in previous studies. Since these chlorophylls are substances that are mainly analyzed in this study, it should be explained in the introduction specifically what advantages they are used for and what characteristics they have.

Reply: Thank you very much for your valuable comments. We have included the following sentences in the Introduction to explain the characteristic of Chl-A and Chl-D (see page 3 bottom to page 4 top of the revised manuscript): “Chl-A, which is a natural chlorophyll-*a* derivative after the peripheral functional group is modified using molecular engineering to obtain a better light-to-photoelectron conversion efficiency, has been applied to organic solar cells, perovskite solar cells, organic-inorganic heterojunction solar cells, and bio-solar cells, exhibiting a significant potential to be a next-generation photoelectronic candidate [3]. Chl-D exhibits ambipolar characteristics owing to the presence of a dicynao-functional group, which makes Chl-D more attractive as both P-type and N-type semiconductors. Furthermore, the application of Chl-A and Chl-D compounds into bio-solar cells inspired by natural photosynthesis has demonstrated good performance [5, 6]. However, the intrinsic excitation dynamics of Chl-A and Chl-D are still unclear [5, 6, 23, 24].” In addition, we clarified how they can be easily synthesized on page 8: “Chl-A and Chl-D are easy to synthesize with high synthetic yields of approximately 80% and 70%, respectively [32-34]. Natural Chl-*a*

was extracted from a commercial cyanobacterium, *Spirulina geitleri*. Demetallation and pyrolysis were then applied to obtain a stable intermediate. Subsequently, hydration and bromination changed the peripheral functional groups of the Chl macrocycle to generate Chl-A or Chl-D”.

2. In the experiment section, the preparation processes of Chl-A and Chl-D solution samples and film samples are different. (a) Why the concentration or amount of Chlorophyll used is not the same? (b) Why are the HQ ratios different? (c) What is the thickness of the spin-coated film samples at 1300 rpm? (d) Why are the device samples spin-coated at 2800 rpm? The HQ ratio of the device samples is very low compared to that of the film samples. Is there a result of film samples with a low HQ ratio or device samples with a high HQ ratio?

Reply: Thank you very much for indicating the important problems. We have addressed each of the queries as listed below.

(a) The concentration of the solution and film samples that we selected for TAS measurement was the best concentration to record the TA signals. The different concentrations and spin-coating speeds of the Chl-A and Chl-D films prepared for the device were the final results after optimizing the photovoltaic performance. Generally, the solution samples with an absorbance of 0.3 at their Q_y maxima in a 2 mm optical path-length cuvette could afford the clearest signal for both TA and GSB signals. For the film samples prepared for the TAS measurement, we observed that the thicker the film, the better the TA signal. We attempted TAS measurements for films with the same thickness as that used for device fabrication. However, recording a TA signal was difficult, because the film prepared for the device was too thin.

(b) We referred to the HQ concentration from the relevant study for the solution samples; thus, the selected concentration of the HQ blended solution was 1.1 mg/ml. According to the Beer–Lambert law, we estimated the concentration of Chl-A. The molar ratio of HQ was approximately 1000 times larger than that of Chl-A in solution samples. Since the electron transfer of the solution samples was diffusion-limited, a large amount of HQ was required to detect its radical scavenging ability. However, Chl-A and HQ were fixed to each other in the film; here, electron transfer between HQ and Chl-A became easier. Therefore, the optimized molar ratio of HQ:Chl-A was 1:1 for

the HQ-blended film samples to record the meaningful TA signal changes induced by the presence of HQ. We have included a related explanation on page 5.

(c) The thickness of the device samples spin-coated at 2800 rpm was 22 nm, as determined using cross-sectional SEM. This SEM image was captured at the Toin University of Yokohama in Japan. However, it was difficult to conduct the same experiment because of the severe effect of COVID-19. Thus, we estimated the thickness of the film spin-coated at 1300 rpm to be approximately 47 nm, assuming that the film thickness was proportional to the spin-coating speed. The device fabrication and estimated thicknesses of the films for TAS measurements are included at the top and middle of page 6, respectively.

Figure R1. Cross-sectional SEM of a Chl-A film.

(d) The spin-speed of 2800 rpm was the final result after device thickness optimization. We also attempted thinner and thicker films of both Chl-A and Chl-D, and their photovoltaic performances were worse than the selected ones (see the solid black line in Figure R2).

Figure R2. Photovoltaic performance of the devices with different Chl-A and Chl-D thicknesses.

(e) As shown in Figure R3, the photovoltaic performance of high-ratio HQ-blended devices (1:1) was worse than that of the pristine device. This is because the overhigh HQ ratio against the Chl-A or Chl-D film may interfere with the self-assembly aggregation of the Chls. Thus, we reduced the ratio of HQ and observed a positive function when HQ was blended with Chl-A at a ratio of 0.5%. We have included this explanation on page 20 of the revised manuscript. We have also included Figure S4 in the Supporting Information.

Figure R3. Photovoltaic performances of the device when the HQ was blended with Chl-A and/or Chl-D at a molar ratio of 1:1, which was the same as the TAS measurement of the film samples.

Table R1. Photovoltaic performances of the Chl-derivative-based bio-solar cells when HQ was blended with Chl-A and/or Chl-D at a molar ratio of 1:1.

Device types	J_{sc} (mA·cm ⁻²)	V_{oc} (V)	FF	PCE (%)
Pristine device	5.07	0.58	0.40	1.18
Chl-A:HQ=1:1	4.96	0.51	0.36	0.91
Chl-D:HQ=1:1	2.63	0.44	0.40	0.46
Both blended at 1:1	2.18	0.42	0.40	1.40

3. Regarding Figure 2, it is necessary to explain the reason or mechanism in which the negative signal does not appear above 500 ns.

Reply: The negative signals of the solution samples over 500 ns still existed but the signal intensity was very weak. If we expand the results of the solution samples to the same intensity scale as the film samples, negative signals can also be observed for the solution samples. Moreover, for the film samples, the strongest negative signals were primarily caused by the scattering of the excitation laser. Therefore, the Q_y bleaching

signal appeared to be strong for the film samples. We have included a corresponding explanation in the middle of page 10 of the revised manuscript.

4. (line 23-24, page 9) *The authors mentioned "singlet lifetime of natural Chls depend on both the solvents and the differences in the molecular peripheral substituents and central metal." The evidence of this statement should be explained in the text, as well as the papers referenced by authors should be cited.*

Reply: We have included the relevant references and the following sentences to the revised manuscript on page 11: “This is because the reported singlet lifetime of natural Chls depends on both the solvents and the differences in the molecular peripheral substituents as well as the central metal [36, 37]. The S_0 to S_1 transition (Q_y band) exhibits high sensitivity to solvent polarity and the π -conjugated chain. The extent of π -conjugation is closely related to the modified peripheral functional group of the tetrapyrrole ring of the Chls.”

5. (line 3-8, page 15 and figure 5 (c)) *The lifetime of carrier species of Chl-D film with or without HQ blending differs more than 10 times, but the triplet lifetime (figure S2) is only doubled. What makes this difference?*

Reply: Thank you very much for indicating this very important point. The doubled triplet lifetime in the Chl-D film did not directly correspond to 10 times increase in the lifetime of the charged carrier species. However, in the case of Chl-A film, the rate of elongation of the carrier lifetime was proportional to that of the triplet lifetime (Table R2). Therefore, a more than 10 times increase in the carrier lifetime in the Chl-D film should be caused by the different carrier generation mechanism from that of the Chl-A film. As exemplified in the solution samples, HQ acted as a cation scavenger for Chl-A. This indicated that HQ is an electron donor to Chl-A. For the Chl-D solution, the radical species were assigned to the anion in this study and HQ donated additional electrons to Chl-D⁻ to generate Chl-D²⁻. Therefore, this may be the reason a more than 10 times elongation of the carrier lifetime was observed for the Chl-D film. If we accept this idea, another question may arise: Why was the carrier lifetime in the Chl-A film increase upon doping with HQ, since HQ should act as a cation scavenger? Here, we

must assume again, that the elongation of the triplet lifetime in the Chl-A film by doping HQ should be the main reason for the elongation of the carrier lifetime. This concept can be supported by considering the rate of elongation of the triple and carrier lifetimes by HQ doping (Table R2). The rate of elongation for the carrier lifetime ($\times 1.28$) was slightly smaller than that of the triplet species ($\times 1.39$), which might have been due to the cation scavenging activity of HQ in the Chl-A film. We have included these explanations and Table 1 in the revised manuscript (bottom of page 16 to the top of page 17).

Table R2. Comparison of the triplet and carrier lifetimes in the Chl-A and Chl-D films with and without HQ.

	Chl-A		Chl-D	
	Without HQ	With HQ	Without HQ	With HQ
Triplet lifetime (ns)	29.3	40.7 ($\times 1.39$ of w/o HQ)	22.6	42.3 ($\times 1.87$ of w/o HQ)
Carrier lifetime (μ s)	1.85	2.38 ($\times 1.28$ of w/o HQ)	0.653	9.05 ($\times 13.9$ of w/o HQ)

6. (Figure 6) Why does the voltage of the device decrease with the higher HQ ratio than 0.5%?

Reply: Thank you again for this important point. The photovoltage of a device is influenced by many factors. As the carrier lifetime increases, the charge recombination among the devices increases if the ratio of HQ is high. However, charge recombination can significantly reduce the voltage. Thus, the photovoltage is reduced accordingly, which is evidenced by the decrease in the FF. Such photocurrent enhancement while photovoltage reduces is normal among devices by doping [44]. We included relevant references to support our thoughts, and a corresponding explanation has been included on page 18 of the revised manuscript.

7. (line 8-16, page 17) Why is the life time balance of donor and acceptor important? And why does the performance deteriorate when they are different? In addition, if the HQ concentration in the Chl-D film samples is very high and the life time balance is

not suitable, it is expected that the life time balance of the device sample can be fitted by adjusting the HQ concentration of Chl-D. The authors should provide the results of a life time balance with Chl-A by adjusting the HQ concentration of Chl-D. Otherwise, it is not convincing that HQ doping of Chl-D simply degrades performance.

Reply: Thank you very much for indicating this essential concern. We apologize that our explanation was not sufficient and caused confusion. The reviewer has observed correctly. Lifetime balance itself is not an essential factor, but we consider that it might be indirectly important to improve performance for the following reasons.

We consider that our solar cell is not a traditional P–N heterojunction-type semiconductor device. In this study, we confirmed that the photo-excited carriers generated in the Chl-A film are cations (holes) and those in Chl-D films are anions (electrons). These carriers are generated at each layer (not at the interface between the Chl-A and Chl-D layers). Therefore, in our solar cell, the electrode adjacent to the Chl-A layer collects electrons, which are discharged from Chl-A, and that adjacent to the Chl-D layer collects holes, which are discharged from Chl-D, while in the typical P–N heterojunction devices, the directions of the carrier flows are opposite. The electrode that is connected to the P-type semiconductor collects holes and that connected to the N-type semiconductor collects electrons. We consider this as a unique feature of our device.

For our device to form an electric circuit, the population of holes in the Chl-A layer and that of electrons in the Chl-D layer should be equilibrated. A balanced carrier lifetime between the donor and acceptor prevents unnecessary charge build-up and consequently reduces the probability of useless charge recombination in either the Chl-A or Chl-D layers. In other words, in charge-unbalanced cells with significant differences in hole- and electron carrier lifetimes, the rapping of the slow carriers results in a build-up of space charges, which induces the reduction of photocurrents by unwanted recombination of the carriers. Therefore, we consider that the device performance might deteriorate when the carrier lifetimes of the donor and acceptor are significantly different.

According to this reviewer's suggestion, we performed additional measurements on the device at different molar ratios of HQ that was doped to the Chl-D layer. The results indicated that the incorporation of HQ into the Chl-D layer always reduced the device

performance (Figure R4); however, HQ doping was effective when it was doped into the Chl-A layer. We have included this result in Figure S5 of the Supporting Information, and a detailed explanation has been included on page 20–21 of the revised manuscript.

Figure R4. *J*-*V* curves of different ratios of HQ doped Chl-D layer-based devices.

8. The authors asserted that adopting HQ can improve the performance of bio-solar cell performance by 20%. However, considering the power conversion efficiency, it does not appear to be significantly improved (from 1.29% to 1.55). The novelty of this paper can be improved by comparing/analyzing the results with other previous studies.

Reply: We are afraid that there are too many to mention, but it is true that the efficiency improvement from pristine 1.29% to HQ-doped 1.55% corresponds to a 20% enhancement; however, there seems to be no significant improvement in the exact value of the power conversion efficiency. Nevertheless, as Reviewer 3 appreciated (see below), our solar cells composed of Chl derivatives exhibit a current density of >5 mA/cm² and monochromatic EQEs >50%, which are considerably good. This is why we consider that the 20% enhancement is meaningful; however, the improvement in the exact value of the power conversion efficiency is not sufficiently high.

We understand that we must correctly claim the novelty of this study as this reviewer has kindly suggested. The aim of this study was to determine the excitation dynamics of Chl-A and Chl-D used for bio-solar cells. The novel finding is that the doping of HQ into the Chl-A layer can improve the photovoltaic performance of our device, the concept of which is derived from our excitation dynamics study. Therefore, we have included the following sentences on page 3 of the revised manuscript: “Previous studies focused primarily on various types of natural Chl and light-harvesting systems without modifications, thus resulting in an insufficient understanding of Chl derivatives that are used as functional materials for solar cells [12-17]. In addition, other studies have investigated the excitation dynamics of tetrapyrrole skeleton-based porphyrin derivatives and their application in solar cells, proving that porphyrin is also a promising photosensitive material for next-generation solar cell applications [18-22]. Moreover, there are no studies in the literature that have investigated the Chl derivatives themselves using ultrafast systems and provided guidance to the next-generation solar cell design.”

Minor issues

1. *Why was all-trans- β -carotene not investigated in chlorophyll solution?*

Reply: The main reason we introduced all-*trans*- β -carotene to the Chl films is to identify the triplet species of Chl-A and Chl-D. As the triplet species of the solution samples have already been identified by introducing nitrogen gas, it is not necessary to introduce all-*trans* β -carotene into the solution samples. In addition, a number of studies concerning natural Chl-*a* and all-*trans* β -carotene blended systems have already been published: Photosynth. Res. 88, 43–50 (2006), Biophys. J. 90, 4145–4154, (2006), Biophys. Chem. 54, 95-107, (1995), etc. The triplet–triplet excitation energy transfer from natural Chl to all-*trans* β -carotene is well established in the field of natural photosynthesis.

2. *In Figures 2 to 5, it is recommended to indicate what each peak is related to in the graph.*

Reply: Thank you for the advice. We did so in the revised Figures 2 and 5.

Responses to the comments from Reviewer #2:

The manuscript entitled, "Redox mediator for enhancing the photovoltaic performance of chlorophyll-based bio-inspired solar cells" reports the influences of redox mediator on bio-inspired solar cell performances. This research provides salient insights in both the characterization and application. Hence, this manuscript can be considered for publication after the mandatory revision on the given aspects:

Reply: Thank you very much for your evaluation and suggestions regarding our work. They do encourage us. We have revised the manuscript accordingly.

1. Language of the manuscript should be improved.

Reply: We have revised the manuscript's language with the aid of native English speakers (English editing company).

2. The introduction part should be precise. So far numerous efforts have been reported on the artificial photosynthesis-based solar cells. In this context, it is important to draw clear lines between other close works and the present work.

Reply: Thank you for your suggestion. We have included additional sentences to clarify the differences and importance of our study. Specifically, we have emphasized the advantages and characteristics of Chl-A and Chl-D. In addition, we have included sentences to clarify the importance of this work at the end of the Introduction. The revised parts are in blue text color.

3. The authors should depict the advantages of bio-inspired solar cells over the conventional solar cells.

Reply: We have included some sentences to describe the advantages of the bio-inspired solar cells on page 3: "These bio-inspired solar cells have significant potential to

achieve high photovoltaic performance by mimicking natural photosynthesis processes that have evolved over billions of years to obtain an optimized light-to-chemical conversion. In addition, these Chl-derivative-based bio-inspired solar cells are biodegradable, which would reduce recycling costs, increase the biological compatibility, and guarantee sustainability.”

4. The synthesis advantages of Chl-A and Chl-D should be illustrated.

Reply: We have included the advantages of their synthesis on page 8: “Chl-A and Chl-D are easy to synthesize with high synthetic yields of approximately 80% and 70%, respectively [32-34]. Natural Chl-*a* was extracted from a commercial cyanobacterium, *Spirulina geitleri*. Demetallation and pyrolysis were then applied to obtain a stable intermediate. Subsequently, hydration and bromination changed the peripheral functional groups of the Chl macrocycle to generate Chl-A or Chl-D”.

5. The significant role of Chl-derivatives on power generation should be detailed.

Reply: Thank you very much for reminding us to describe this important item. We have explained the function of Chl derivatives on photocurrent generation on page 17 of the revised manuscript: “Chl-A and Chl-D act as light-harvesting materials in the 350–700 nm wavelength region, which resembles the Z-scheme of natural photosynthesis. Therefore, we called our device a bio-inspired solar cell. The incident photons of Chl-A and Chl-D generate photo-excited carriers. A cation radical (hole) is produced in the Chl-A layer, and the electron is transferred to the cathode, while the anion radical (electron) is produced in Chl-D layer, and the hole is transferred to the anode. These carriers (hole from Chl-A and electron from Chl-D) are recombined at the interface between the Chl-A and Chl-D layers, and a photocurrent is finally generated by this device under illumination.”

6. The reasons for high power generation of a specific Chl derivative should be depicted.

Reply: On page 4, we have included the related sentences to emphasize the characteristics of the specific Chl derivative to attain a better photovoltaic performance: “Generally, strong light absorption, high carrier mobility, and long excitation lifetime

are essential for Chl derivatives to achieve high photovoltaic performance. This is because the strong light-absorption ability can enable a device to harvest more photons, while the high carrier mobility and long excitation lifetimes contribute to charge separation and transportation [25].”

7. *Have the authors optimized thickness of variant layers in solar cell fabrication? If so, detail the same.*

Reply: We optimized the thickness of each layer during solar cell fabrication and the results are shown in Figure R5. The red line *J-V* curve represents the device with the highest photovoltaic performance. This means that 2800 rpm was the most suitable spin-coating speed. The film thickness was 22 nm. These results are shown in Figure S3 of the Supporting Information.

Figure R5. Effect of the film thickness of Chl-A (a) and Chl-D (b) on the final photovoltaic performance of the Z-scheme photosynthesis-inspired devices.

8. *The mechanism involved in the power generation should be schematically illustrated.*

Reply: Thank you very much for your advice. We have included the explanation of the operating mechanisms of our bio-inspired solar cells on page 21: “Electron and hole pairs (excitons) are generated from triplet Chl-A and Chl-D after photoexcitation. The generated holes of Chl-A are recombined with the generated electrons of Chl-D at the interface of the Chl-A and Chl-D layers. In contrast, the electrons discharged from Chl-A are extracted by the ZnO electron transporter and collected by the ITO cathode, while

the holes discharged from Chl-D migrate to the Ag anode through the MoO₃ electron blocker when the device is operating”.

Responses to the comments from Reviewer #3:

1. The authors provide compelling evidence that chlorophyll A and D have long lived triplet states that are quenched by O₂ in solution and in thin films. Many macrocyclic organic excited states, such as porphyrins, show similar behavior. This reviewer does not know how novel this observation is for chlorophylls.

Reply: We apologize that we cannot fully explain the novelty of our work to this reviewer; however, Reviewer 2 favors this research as it provides salient insights into both the characterization and application, and recommends the publication of this paper.

The main objective of this study was to clarify the excitation dynamics (singlet states, triplet states, and radicals (carriers in thin solid films) of Chl derivatives and to determine the operating mechanisms of Chl-derivative-inspired bio-solar cells. The introduction of nitrogen gas to the solution sample was a part of this study to identify the triplet species. In addition, no similar study has been conducted on the excitation dynamics of the Chl derivatives that are used for solar cells. Therefore, we consider that this study fills the gap between the intrinsic dynamic study of functional Chl derivatives and solar cell device applications. This is the novelty of our research.

We did not directly apply O₂ to either the solution samples or film samples to examine the triplet species produced from Chl-A and Chl-D. N₂ gas was introduced only to the solution samples, while all-*trans* β-carotene was blended with the pristine Chl-A and Chl-D films to identify the triplet species. The application of N₂ gas to the solution samples reduced the contamination of O₂; hence, the triplet lifetime would exhibit an increase. We could not apply N₂ gas to the film samples because the diffusion of N₂ gas in the thin solid films was not sufficient to indicate the lifetime increase of the triplet species. Therefore, we blended all-*trans* β-carotene to the film samples, since the triplet excited state of all-*trans*-β-carotene can only be produced through triple-triplet excitation energy transfer from the Chl derivatives. If the production of

triplet β -carotene is observed, this provides good evidence that the triplet excited states of Chl derivatives are produced.

2. The triplet states are competent of oxidation of hydroquinone (HQ) and/or reduction of beta-carotene. Energy transfer to beta-carotene was also observed. It was unclear whether the authors believe that the triplets annihilate (i.e. disproportionate) and then the products are scavenged or whether the triplets undergo direct electron transfer with HQ and B-car. The 'lifetimes' are longer after these electron transfer events than for the triplets alone.

Reply: We do not consider that the triplet species of Chls can oxidize HQ or reduce all-*trans* β -carotene; however, the reviewer suspects whether the triplets are annihilated (i.e. disproportionated) and then the products are scavenged or the triplets undergo direct electron transfer with HQ and β -carotene. However, there is no electron transfer to all-*trans* β -carotene, as it just acts as a triplet energy acceptor; hence, we call it the triplet scavenger. In view of this, we are not very certain about the exact mechanisms by which the radical species of Chl-A and Chl-D have been generated because this is the first exploration of this phenomenon. However, our results, based on a sequential analysis model and observed kinetics, clearly indicate that the radical species of Chl-A and Chl-D originate from the triplet species.

3. Solar cells comprised of these materials show $> 5 \text{ mA/cm}^2$ of current density and monochromatic EQEs $> 50\%$ which is quite good.

Reply: Thank you very much for your high evaluation. Our *J-V* results could closely match with the EQE results.

4. Overall I found this paper to be confusing to read. Calling the HQ and B-Car scavengers and referring to their charge rather than calling them acceptors and donors and referring to whether they were reduced and oxidized was unfortunate. Figure 7 looks more like a diagram for a semiconductor than for molecular dyes. Also referring to 'carriers' was vague as in some cases this was referring to chlorophyll triplets and in other cases charge-separated states. If there is a lot of triple-triple annihilation then the charge separated states will likely live longer and this is not a surprise. The true

lifetime of organic triplets remains contentious in many cases as they are so long-lived that their lifetimes are often limited by quenching with impurities.

Reply: We apologize for causing confusion. However, throughout this manuscript, HQ is termed a radical cation scavenger, while β -carotene is a triplet energy scavenger. The aim of this study was to clarify the excitation dynamics of Chl-A and Chl-D both in the solution and film states, and to subsequently determine the operating principle of the Chl-A- and Chl-D-based natural Z-scheme photosynthesis inspired bio-solar cells. After this investigation, we concluded that our hypothesis was correct. We considered it necessary to propose the exact operating mechanisms of the relevant bio-solar cell device with our new finding; therefore, we combined the device energy alignments of each layer and the excitation dynamics of the Chls and attempt to explain using the diagrams in Figure 7. We have included the relevant sentences on page 21 of the revised manuscript. In the manuscript, we assume that the carrier species that carry electrons or holes in the film originate from the radical species that were observed in the solution. We did not mix the word “carriers” with the charge-separated states (please see the response to Reviewer 1, query #7). In addition, we must emphasize that we did not observe any triplet–triplet annihilation in this study because all the kinetics could be perfectly analyzed based on a sequential model. In this study, we clearly observed that the carrier and radical species were produced from the triplet species according to our sequential model.

REVIEWERS' COMMENTS:

Reviewer #1 (Remarks to the Author):

The comments and criticisms raised by the reviewer have been all properly addressed by the author responses and the revised manuscript. I believe that the manuscript is now significantly improved, and I think it can be accepted for publication now.

Reviewer #2 (Remarks to the Author):

As the authors improved the quality of this article under the limelight of reviewer's comments, this manuscript can be accepted for publication.

Response to the editor's comments:

As you will see, all Reviewers mention that the manuscript would benefit from a clearer presentation of the advance your work represents over closely related prior work. In addition, Reviewer #1 notes that the choice of chlorophylls and the preparation of solution and thin films is currently unclear, and that the device performance such as voltage decrease need to be clarified. The claim that HQ doping of Chl-D degrades performance needs to be further substantiated with data.

Reply: We have made substantial revisions to the manuscript. The revised parts are in blue text color in the revised manuscript.

(1) We emphasized the importance of this study and its differences from the previous studies. The corresponding revisions are provided in the Introduction of the manuscript.

(2) The choice of Chls and the preparation of the solution and thin films, which Reviewer 1 mentioned, have been answered specifically in the main text (please see the reply to Reviewer 1).

(3) The reason for the changes in the device performance after HQ doping with different ratios has been explained using the data obtained from additional experiments.

(4) The photovoltaic performances of the HQ-doped Chl-D devices have been included and discussed in the manuscript, and the relevant results are depicted in Figure S5 of the Supporting Information.

Responses to Reviewers' comments:**Responses to the comments from Reviewer #1:**

The manuscript "Redox mediator for enhancing the photovoltaic performance of chlorophyll-based bio-inspired solar cells" by Shengnan Duan et al. covers the construction of bio-solar cells and evaluation of their performance through the analysis of chlorophyll derivatives including Chl-A and Chl-D. In particular, the

excitation dynamics were investigated in the film and solution forms of Chl-A and Chl-D, and in the process, hydroquinone (HQ), which is a cation radical scavenger and/or anion radical donor, and all-trans- β -carotene, which is a triplet scavenger, were added to measure the life time of radical species and carrier species. Based on the results of the life time of radical and carrier species, bio-inspired solar cells with different HQ ratios to the Chl-A were constructed and verified. Overall, the process of analyzing chlorophyll derivatives, constructing cells, and evaluating their performance was well organized, but with some lack of explanation, some questions still exist. For these reasons, the manuscript cannot be acceptable in the present form. I would suggest the authors consider the points raised below.

Reply: Thank you very much for your positive evaluation of our study. We have made revisions according to your suggestions.

Major issues

1. It is only mentioned that Chl-A and Chl-D were used in previous studies. Since these chlorophylls are substances that are mainly analyzed in this study, it should be explained in the introduction specifically what advantages they are used for and what characteristics they have.

Reply: Thank you very much for your valuable comments. We have included the following sentences in the Introduction to explain the characteristic of Chl-A and Chl-D (see page 3 bottom to page 4 top of the revised manuscript): “Chl-A, which is a natural chlorophyll-*a* derivative after the peripheral functional group is modified using molecular engineering to obtain a better light-to-photoelectron conversion efficiency, has been applied to organic solar cells, perovskite solar cells, organic-inorganic heterojunction solar cells, and bio-solar cells, exhibiting a significant potential to be a next-generation photoelectronic candidate [3]. Chl-D exhibits ambipolar characteristics owing to the presence of a dicynao-functional group, which makes Chl-D more attractive as both P-type and N-type semiconductors. Furthermore, the application of Chl-A and Chl-D compounds into bio-solar cells inspired by natural photosynthesis has demonstrated good performance [5, 6]. However, the intrinsic excitation dynamics of Chl-A and Chl-D are still unclear [5, 6, 23, 24].” In addition,

we clarified how they can be easily synthesized on page 8: “Chl-A and Chl-D are easy to synthesize with high synthetic yields of approximately 80% and 70%, respectively [32-34]. Natural Chl-*a* was extracted from a commercial cyanobacterium, *Spirulina geitleri*. Demetallation and pyrolysis were then applied to obtain a stable intermediate. Subsequently, hydration and bromination changed the peripheral functional groups of the Chl macrocycle to generate Chl-A or Chl-D”.

2. In the experiment section, the preparation processes of Chl-A and Chl-D solution samples and film samples are different. (a) Why the concentration or amount of Chlorophyll used is not the same? (b) Why are the HQ ratios different? (c) What is the thickness of the spin-coated film samples at 1300 rpm? (d) Why are the device samples spin-coated at 2800 rpm? The HQ ratio of the device samples is very low compared to that of the film samples. Is there a result of film samples with a low HQ ratio or device samples with a high HQ ratio?

Reply: Thank you very much for indicating the important problems. We have addressed each of the queries as listed below.

(a) The concentration of the solution and film samples that we selected for TAS measurement was the best concentration to record the TA signals. The different concentrations and spin-coating speeds of the Chl-A and Chl-D films prepared for the device were the final results after optimizing the photovoltaic performance. Generally, the solution samples with an absorbance of 0.3 at their Q_y maxima in a 2 mm optical path-length cuvette could afford the clearest signal for both TA and GSB signals. For the film samples prepared for the TAS measurement, we observed that the thicker the film, the better the TA signal. We attempted TAS measurements for films with the same thickness as that used for device fabrication. However, recording a TA signal was difficult, because the film prepared for the device was too thin.

(b) We referred to the HQ concentration from the relevant study for the solution samples; thus, the selected concentration of the HQ blended solution was 1.1 mg/ml. According to the Beer–Lambert law, we estimated the concentration of Chl-A. The molar ratio of HQ was approximately 1000 times larger than that of Chl-A in solution samples. Since the electron transfer of the solution samples was diffusion-limited, a large amount of HQ was required to detect its radical scavenging ability. However,

Chl-A and HQ were fixed to each other in the film; here, electron transfer between HQ and Chl-A became easier. Therefore, the optimized molar ratio of HQ:Chl-A was 1:1 for the HQ-blended film samples to record the meaningful TA signal changes induced by the presence of HQ. We have included a related explanation on page 5.

(c) The thickness of the device samples spin-coated at 2800 rpm was 22 nm, as determined using cross-sectional SEM. This SEM image was captured at the Toim University of Yokohama in Japan. However, it was difficult to conduct the same experiment because of the severe effect of COVID-19. Thus, we estimated the thickness of the film spin-coated at 1300 rpm to be approximately 47 nm, assuming that the film thickness was proportional to the spin-coating speed. The device fabrication and estimated thicknesses of the films for TAS measurements are included at the top and middle of page 6, respectively.

Figure R1. Cross-sectional SEM of a Chl-A film.

(d) The spin-speed of 2800 rpm was the final result after device thickness optimization. We also attempted thinner and thicker films of both Chl-A and Chl-D, and their photovoltaic performances were worse than the selected ones (see the solid black line in Figure R2).

Figure R2. Photovoltaic performance of the devices with different Chl-A and Chl-D thicknesses.

(e) As shown in Figure R3, the photovoltaic performance of high-ratio HQ-blended devices (1:1) was worse than that of the pristine device. This is because the overhigh HQ ratio against the Chl-A or Chl-D film may interfere with the self-assembly aggregation of the Chls. Thus, we reduced the ratio of HQ and observed a positive function when HQ was blended with Chl-A at a ratio of 0.5%. We have included this explanation on page 20 of the revised manuscript. We have also included Figure S4 in the Supporting Information.

Figure R3. Photovoltaic performances of the device when the HQ was blended with Chl-A and/or Chl-D at a molar ratio of 1:1, which was the same as the TAS measurement of the film samples.

Table R1. Photovoltaic performances of the Chl-derivative-based bio-solar cells when HQ was blended with Chl-A and/or Chl-D at a molar ratio of 1:1.

Device types	J_{sc} (mA \square cm ⁻²)	V_{oc} (V)	FF	PCE (%)
Pristine device	5.07	0.58	0.40	1.18
Chl-A:HQ=1:1	4.96	0.51	0.36	0.91
Chl-D:HQ=1:1	2.63	0.44	0.40	0.46
Both blended at 1:1	2.18	0.42	0.40	1.40

3. Regarding Figure 2, it is necessary to explain the reason or mechanism in which the negative signal does not appear above 500 ns.

Reply: The negative signals of the solution samples over 500 ns still existed but the signal intensity was very weak. If we expand the results of the solution samples to the same intensity scale as the film samples, negative signals can also be observed for the solution samples. Moreover, for the film samples, the strongest negative signals were primarily caused by the scattering of the excitation laser. Therefore, the Q_y bleaching

signal appeared to be strong for the film samples. We have included a corresponding explanation in the middle of page 10 of the revised manuscript.

4. (line 23-24, page 9) *The authors mentioned "singlet lifetime of natural Chls depend on both the solvents and the differences in the molecular peripheral substituents and central metal." The evidence of this statement should be explained in the text, as well as the papers referenced by authors should be cited.*

Reply: We have included the relevant references and the following sentences to the revised manuscript on page 11: "This is because the reported singlet lifetime of natural Chls depends on both the solvents and the differences in the molecular peripheral substituents as well as the central metal [36, 37]. The S_0 to S_1 transition (Q_y band) exhibits high sensitivity to solvent polarity and the π -conjugated chain. The extent of π -conjugation is closely related to the modified peripheral functional group of the tetrapyrrole ring of the Chls."

5. (line 3-8, page 15 and figure 5 (c)) *The lifetime of carrier species of Chl-D film with or without HQ blending differs more than 10 times, but the triplet lifetime (figure S2) is only doubled. What makes this difference?*

Reply: Thank you very much for indicating this very important point. The doubled triplet lifetime in the Chl-D film did not directly correspond to 10 times increase in the lifetime of the charged carrier species. However, in the case of Chl-A film, the rate of elongation of the carrier lifetime was proportional to that of the triplet lifetime (Table R2). Therefore, a more than 10 times increase in the carrier lifetime in the Chl-D film should be caused by the different carrier generation mechanism from that of the Chl-A film. As exemplified in the solution samples, HQ acted as a cation scavenger for Chl-A. This indicated that HQ is an electron donor to Chl-A. For the Chl-D solution, the radical species were assigned to the anion in this study and HQ donated additional electrons to Chl-D⁻ to generate Chl-D²⁻. Therefore, this may be the reason a more than 10 times elongation of the carrier lifetime was observed for the Chl-D film. If we accept this idea, another question may arise: Why was the carrier lifetime in the Chl-A film increase upon doping with HQ, since HQ should act as a

cation scavenger? Here, we must assume again, that the elongation of the triplet lifetime in the Chl-A film by doping HQ should be the main reason for the elongation of the carrier lifetime. This concept can be supported by considering the rate of elongation of the triple and carrier lifetimes by HQ doping (Table R2). The rate of elongation for the carrier lifetime ($\times 1.28$) was slightly smaller than that of the triplet species ($\times 1.39$), which might have been due to the cation scavenging activity of HQ in the Chl-A film. We have included these explanations and Table 1 in the revised manuscript (bottom of page 16 to the top of page 17).

Table R2. Comparison of the triplet and carrier lifetimes in the Chl-A and Chl-D films with and without HQ.

	Chl-A		Chl-D	
	Without HQ	With HQ	Without HQ	With HQ
Triplet lifetime (ns)	29.3	40.7 ($\times 1.39$ of w/o HQ)	22.6	42.3 ($\times 1.87$ of w/o HQ)
Carrier lifetime (μ s)	1.85	2.38 ($\times 1.28$ of w/o HQ)	0.653	9.05 ($\times 13.9$ of w/o HQ)

6. (Figure 6) Why does the voltage of the device decrease with the higher HQ ratio than 0.5%?

Reply: Thank you again for this important point. The photovoltage of a device is influenced by many factors. As the carrier lifetime increases, the charge recombination among the devices increases if the ratio of HQ is high. However, charge recombination can significantly reduce the voltage. Thus, the photovoltage is reduced accordingly, which is evidenced by the decrease in the FF. Such photocurrent enhancement while photovoltage reduces is normal among devices by doping [44]. We included relevant references to support our thoughts, and a corresponding explanation has been included on page 18 of the revised manuscript.

7. (line 8-16, page 17) Why is the life time balance of donor and acceptor important? And why does the performance deteriorate when they are different? In addition, if the HQ concentration in the Chl-D film samples is very high and the life time balance is

not suitable, it is expected that the life time balance of the device sample can be fitted by adjusting the HQ concentration of Chl-D. The authors should provide the results of a life time balance with Chl-A by adjusting the HQ concentration of Chl-D. Otherwise, it is not convincing that HQ doping of Chl-D simply degrades performance.

Reply: Thank you very much for indicating this essential concern. We apologize that our explanation was not sufficient and caused confusion. The reviewer has observed correctly. Lifetime balance itself is not an essential factor, but we consider that it might be indirectly important to improve performance for the following reasons.

We consider that our solar cell is not a traditional P–N heterojunction-type semiconductor device. In this study, we confirmed that the photo-excited carriers generated in the Chl-A film are cations (holes) and those in Chl-D films are anions (electrons). These carriers are generated at each layer (not at the interface between the Chl-A and Chl-D layers). Therefore, in our solar cell, the electrode adjacent to the Chl-A layer collects electrons, which are discharged from Chl-A, and that adjacent to the Chl-D layer collects holes, which are discharged from Chl-D, while in the typical P–N heterojunction devices, the directions of the carrier flows are opposite. The electrode that is connected to the P-type semiconductor collects holes and that connected to the N-type semiconductor collects electrons. We consider this as a unique feature of our device.

For our device to form an electric circuit, the population of holes in the Chl-A layer and that of electrons in the Chl-D layer should be equilibrated. A balanced carrier lifetime between the donor and acceptor prevents unnecessary charge build-up and consequently reduces the probability of useless charge recombination in either the Chl-A or Chl-D layers. In other words, in charge-unbalanced cells with significant differences in hole- and electron carrier lifetimes, the rapping of the slow carriers results in a build-up of space charges, which induces the reduction of photocurrents by unwanted recombination of the carriers. Therefore, we consider that the device performance might deteriorate when the carrier lifetimes of the donor and acceptor are significantly different.

According to this reviewer's suggestion, we performed additional measurements on the device at different molar ratios of HQ that was doped to the Chl-D layer. The

results indicated that the incorporation of HQ into the Chl-D layer always reduced the device performance (Figure R4); however, HQ doping was effective when it was doped into the Chl-A layer. We have included this result in Figure S5 of the Supporting Information, and a detailed explanation has been included on page 20–21 of the revised manuscript.

Figure R4. *J-V* curves of different ratios of HQ doped Chl-D layer-based devices.

8. *The authors asserted that adopting HQ can improve the performance of bio-solar cell performance by 20%. However, considering the power conversion efficiency, it does not appear to be significantly improved (from 1.29% to 1.55). The novelty of this paper can be improved by comparing/analyzing the results with other previous studies.*

Reply: We are afraid that there are too many to mention, but it is true that the efficiency improvement from pristine 1.29% to HQ-doped 1.55% corresponds to a 20% enhancement; however, there seems to be no significant improvement in the exact value of the power conversion efficiency. Nevertheless, as Reviewer 3 appreciated (see below), our solar cells composed of Chl derivatives exhibit a current density of $>5 \text{ mA/cm}^2$ and monochromatic EQEs $>50\%$, which are considerably good. This is

why we consider that the 20% enhancement is meaningful; however, the improvement in the exact value of the power conversion efficiency is not sufficiently high.

We understand that we must correctly claim the novelty of this study as this reviewer has kindly suggested. The aim of this study was to determine the excitation dynamics of Chl-A and Chl-D used for bio-solar cells. The novel finding is that the doping of HQ into the Chl-A layer can improve the photovoltaic performance of our device, the concept of which is derived from our excitation dynamics study. Therefore, we have included the following sentences on page 3 of the revised manuscript: “Previous studies focused primarily on various types of natural Chl and light-harvesting systems without modifications, thus resulting in an insufficient understanding of Chl derivatives that are used as functional materials for solar cells [12-17]. In addition, other studies have investigated the excitation dynamics of tetrapyrrole skeleton-based porphyrin derivatives and their application in solar cells, proving that porphyrin is also a promising photosensitive material for next-generation solar cell applications [18-22]. Moreover, there are no studies in the literature that have investigated the Chl derivatives themselves using ultrafast systems and provided guidance to the next-generation solar cell design.”

Minor issues

1. Why was all-trans- β -carotene not investigated in chlorophyll solution?

Reply: The main reason we introduced all-*trans*- β -carotene to the Chl films is to identify the triplet species of Chl-A and Chl-D. As the triplet species of the solution samples have already been identified by introducing nitrogen gas, it is not necessary to introduce all-*trans* β -carotene into the solution samples. In addition, a number of studies concerning natural Chl-*a* and all-*trans* β -carotene blended systems have already been published: Photosynth. Res. 88, 43–50 (2006), Biophys. J. 90, 4145–4154, (2006), Biophys. Chem. 54, 95-107, (1995), etc. The triplet–triplet excitation energy transfer from natural Chl to all-*trans* β -carotene is well established in the field of natural photosynthesis.

2. In Figures 2 to 5, it is recommended to indicate what each peak is related to in the graph.

Reply: Thank you for the advice. We did so in the revised Figures 2 and 5.

Responses to the comments from Reviewer #2:

The manuscript entitled, "Redox mediator for enhancing the photovoltaic performance of chlorophyll-based bio-inspired solar cells" reports the influences of redox mediator on bio-inspired solar cell performances. This research provides salient insights in both the characterization and application. Hence, this manuscript can be considered for publication after the mandatory revision on the given aspects:

Reply: Thank you very much for your evaluation and suggestions regarding our work. They do encourage us. We have revised the manuscript accordingly.

1. *Language of the manuscript should be improved.*

Reply: We have revised the manuscript's language with the aid of native English speakers (English editing company).

2. *The introduction part should be precise. So far numerous efforts have been reported on the artificial photosynthesis-based solar cells. In this context, it is important to draw clear lines between other close works and the present work.*

Reply: Thank you for your suggestion. We have included additional sentences to clarify the differences and importance of our study. Specifically, we have emphasized the advantages and characteristics of Chl-A and Chl-D. In addition, we have included sentences to clarify the importance of this work at the end of the Introduction. The revised parts are in blue text color.

3. *The authors should depict the advantages of bio-inspired solar cells over the conventional solar cells.*

Reply: We have included some sentences to describe the advantages of the bio-inspired solar cells on page 3: “These bio-inspired solar cells have significant potential to achieve high photovoltaic performance by mimicking natural photosynthesis processes that have evolved over billions of years to obtain an optimized light-to-chemical conversion. In addition, these Chl-derivative-based bio-inspired solar cells are bio-degradable, which would reduce recycling costs, increase the biological compatibility, and guarantee sustainability.”

4. The synthesis advantages of Chl-A and Chl-D should be illustrated.

Reply: We have included the advantages of their synthesis on page 8: “Chl-A and Chl-D are easy to synthesize with high synthetic yields of approximately 80% and 70%, respectively [32-34]. Natural Chl-*a* was extracted from a commercial cyanobacterium, *Spirulina geitleri*. Demetallation and pyrolysis were then applied to obtain a stable intermediate. Subsequently, hydration and bromination changed the peripheral functional groups of the Chl macrocycle to generate Chl-A or Chl-D”.

5. The significant role of Chl-derivatives on power generation should be detailed.

Reply: Thank you very much for reminding us to describe this important item. We have explained the function of Chl derivatives on photocurrent generation on page 17 of the revised manuscript: “Chl-A and Chl-D act as light-harvesting materials in the 350–700 nm wavelength region, which resembles the Z-scheme of natural photosynthesis. Therefore, we called our device a bio-inspired solar cell. The incident photons of Chl-A and Chl-D generate photo-excited carriers. A cation radical (hole) is produced in the Chl-A layer, and the electron is transferred to the cathode, while the anion radical (electron) is produced in Chl-D layer, and the hole is transferred to the anode. These carriers (hole from Chl-A and electron from Chl-D) are recombined at the interface between the Chl-A and Chl-D layers, and a photocurrent is finally generated by this device under illumination.”

6. The reasons for high power generation of a specific Chl derivative should be depicted.

Reply: On page 4, we have included the related sentences to emphasize the characteristics of the specific Chl derivative to attain a better photovoltaic performance: “Generally, strong light absorption, high carrier mobility, and long excitation lifetime are essential for Chl derivatives to achieve high photovoltaic performance. This is because the strong light-absorption ability can enable a device to harvest more photons, while the high carrier mobility and long excitation lifetimes contribute to charge separation and transportation [25].”

7. *Have the authors optimized thickness of variant layers in solar cell fabrication? If so, detail the same.*

Reply: We optimized the thickness of each layer during solar cell fabrication and the results are shown in Figure R5. The red line J - V curve represents the device with the highest photovoltaic performance. This means that 2800 rpm was the most suitable spin-coating speed. The film thickness was 22 nm. These results are shown in Figure S3 of the Supporting Information.

Figure R5. Effect of the film thickness of Chl-A (a) and Chl-D (b) on the final photovoltaic performance of the Z-scheme photosynthesis-inspired devices.

8. *The mechanism involved in the power generation should be schematically illustrated.*

Reply: Thank you very much for your advice. We have included the explanation of the operating mechanisms of our bio-inspired solar cells on page 21: “Electron and

hole pairs (excitons) are generated from triplet Chl-A and Chl-D after photoexcitation. The generated holes of Chl-A are recombined with the generated electrons of Chl-D at the interface of the Chl-A and Chl-D layers. In contrast, the electrons discharged from Chl-A are extracted by the ZnO electron transporter and collected by the ITO cathode, while the holes discharged from Chl-D migrate to the Ag anode through the MoO₃ electron blocker when the device is operating”.

Responses to the comments from Reviewer #3:

1. The authors provide compelling evidence that chlorophyll A and D have long lived triplet states that are quenched by O₂ in solution and in thin films. Many macrocyclic organic excited states, such as porphyrins, show similar behavior. This reviewer does not know how novel this observation is for chlorophylls.

Reply: We apologize that we cannot fully explain the novelty of our work to this reviewer; however, Reviewer 2 favors this research as it provides salient insights into both the characterization and application, and recommends the publication of this paper.

The main objective of this study was to clarify the excitation dynamics (singlet states, triplet states, and radicals (carriers in thin solid films) of Chl derivatives and to determine the operating mechanisms of Chl-derivative-inspired bio-solar cells. The introduction of nitrogen gas to the solution sample was a part of this study to identify the triplet species. In addition, no similar study has been conducted on the excitation dynamics of the Chl derivatives that are used for solar cells. Therefore, we consider that this study fills the gap between the intrinsic dynamic study of functional Chl derivatives and solar cell device applications. This is the novelty of our research.

We did not directly apply O₂ to either the solution samples or film samples to examine the triplet species produced from Chl-A and Chl-D. N₂ gas was introduced only to the solution samples, while all-*trans* β-carotene was blended with the pristine Chl-A and Chl-D films to identify the triplet species. The application of N₂ gas to the solution samples reduced the contamination of O₂; hence, the triplet lifetime would exhibit an increase. We could not apply N₂ gas to the film samples because the diffusion of N₂ gas in the thin solid films was not sufficient to indicate the lifetime

increase of the triplet species. Therefore, we blended all-*trans* β -carotene to the film samples, since the triplet excited state of all-*trans*- β -carotene can only be produced through triple–triplet excitation energy transfer from the Chl derivatives. If the production of triplet β -carotene is observed, this provides good evidence that the triplet excited states of Chl derivatives are produced.

2. The triplet states are competent of oxidation of hydroquinone (HQ) and/or reduction of beta-carotene. Energy transfer to beta-carotene was also observed. It was unclear whether the authors believe that the triplets annihilate (i.e. disproportionate) and then the products are scavenged or whether the triplets undergo direct electron transfer with HQ and B-car. The 'lifetimes' are longer after these electron transfer events than for the triplets alone.

Reply: We do not consider that the triplet species of Chls can oxidize HQ or reduce all-*trans* β -carotene; however, the reviewer suspects whether the triplets are annihilated (i.e. disproportionated) and then the products are scavenged or the triplets undergo direct electron transfer with HQ and β -carotene. However, there is no electron transfer to all-*trans* β -carotene, as it just acts as a triplet energy acceptor; hence, we call it the triplet scavenger. In view of this, we are not very certain about the exact mechanisms by which the radical species of Chl-A and Chl-D have been generated because this is the first exploration of this phenomenon. However, our results, based on a sequential analysis model and observed kinetics, clearly indicate that the radical species of Chl-A and Chl-D originate from the triplet species.

3. Solar cells comprised of these materials show > 5 mA/cm² of current density and monochromatic EQEs > 50% which is quite good.

Reply: Thank you very much for your high evaluation. Our *J-V* results could closely match with the EQE results.

4. Overall I found this paper to be confusing to read. Calling the HQ and B-Car scavengers and referring to their charge rather than calling them acceptors and donors and referring to whether they were reduced and oxidized was unfortunate. Figure 7 looks more like a diagram for a semiconductor than for molecular dyes. Also

referring to 'carriers' was vague as in some cases this was referring to chlorophyll triplets and in other cases charge-separated states. If there is a lot of triple-triplet annihilation then the charge separated states will likely live longer and this is not a surprise. The true lifetime of organic triplets remains contentious in many cases as they are so long-lived that their lifetimes are often limited by quenching with impurities.

Reply: We apologize for causing confusion. However, throughout this manuscript, HQ is termed a radical cation scavenger, while β -carotene is a triplet energy scavenger. The aim of this study was to clarify the excitation dynamics of Chl-A and Chl-D both in the solution and film states, and to subsequently determine the operating principle of the Chl-A- and Chl-D-based natural Z-scheme photosynthesis inspired bio-solar cells. After this investigation, we concluded that our hypothesis was correct. We considered it necessary to propose the exact operating mechanisms of the relevant bio-solar cell device with our new finding; therefore, we combined the device energy alignments of each layer and the excitation dynamics of the Chls and attempt to explain using the diagrams in Figure 7. We have included the relevant sentences on page 21 of the revised manuscript. In the manuscript, we assume that the carrier species that carry electrons or holes in the film originate from the radical species that were observed in the solution. We did not mix the word “carriers” with the charge-separated states (please see the response to Reviewer 1, query #7). In addition, we must emphasize that we did not observe any triplet–triplet annihilation in this study because all the kinetics could be perfectly analyzed based on a sequential model. In this study, we clearly observed that the carrier and radical species were produced from the triplet species according to our sequential model.